# A lipoprotein allosterically activates the CwlD amidase during *Clostridioides difficile* spore formation

Carolina Alves Feliciano[1☉¤a], Brian E. Eckenroth[2☉], Oscar R. Diaz[1¤b], Sylvie Doublié[2], Aimee Shen[1]*

**1** Department of Molecular Biology and Microbiology, Tufts University School of Medicine, Boston, Massachusetts, United States of America, **2** Department of Microbiology and Molecular Genetics, University of Vermont, Burlington, Vermont, United States of America

☉ These authors contributed equally to this work.
¤a Current address: Instituto de Tecnologia Química e Biológica Antonio Xavier, Universidade Nova de Lisboa, Oeiras, Portugal
¤b Current address: Department of Microbiology & Immunology, Stanford School of Medicine, Stanford, California, United States of America
* aimee.shen@tufts.edu

**Data Availability Statement:** The crystallographic model and associated structure factors are available from the Protein Data Bank with PDBID

## Abstract

Spore-forming pathogens like *Clostridioides difficile* depend on germination to initiate infection. During gemination, spores must degrade their cortex layer, which is a thick, protective layer of modified peptidoglycan. Cortex degradation depends on the presence of the spore-specific peptidoglycan modification, muramic-$\partial$-lactam (MAL), which is specifically recognized by cortex lytic enzymes. In *C. difficile*, MAL production depends on the CwlD amidase and its binding partner, the GerS lipoprotein. To gain insight into how GerS regulates CwlD activity, we solved the crystal structure of the CwlD:GerS complex. In this structure, a GerS homodimer is bound to two CwlD monomers such that the CwlD active sites are exposed. Although CwlD structurally resembles amidase_3 family members, we found that CwlD does not bind $Zn^{2+}$ stably on its own, unlike previously characterized amidase_3 enzymes. Instead, GerS binding to CwlD promotes CwlD binding to $Zn^{2+}$, which is required for its catalytic mechanism. Thus, in determining the first structure of an amidase bound to its regulator, we reveal stabilization of $Zn^{2+}$ co-factor binding as a novel mechanism for regulating bacterial amidase activity. Our results further suggest that allosteric regulation by binding partners may be a more widespread mode for regulating bacterial amidase activity than previously thought.

## Author summary

Spore germination is essential for many spore-forming pathogens to initiate infection. In order for spores to germinate, they must degrade a thick, protective layer of cell wall known as the cortex. The enzymes that digest this layer selectively recognize the spore-specific cell wall modification, muramic-$\partial$-lactam (MAL). MAL is made in part through

7RAG. All other relevant data are within the manuscript and its Supporting Information files.

**Funding:** Research in this manuscript was funded by R01GM108684 and R01GM140361 from the National Institutes of General Medical Sciences (NIGMS) to A.S. A.S. is a Burroughs Wellcome Investigator in the Pathogenesis of Infectious Disease supported by the Burroughs Wellcome Fund. B.E.E. is supported by NIH grant R50 CA233185. Single crystal X-ray diffraction experiments were conducted at GM/CA@APS which has been funded in whole or in part with Federal funds from the National Cancer Institute (ACB-12002) and the National Institute of General Medical Sciences (AGM-12006). This research used resources of the Advanced Photon Source, a U.S. Department of Energy (DOE) Office of Science User Facility operated for the DOE Office of Science by Argonne National Laboratory under Contract No. DE-AC02-06CH11357. O.D. was funded by R25 GM066567-15 from the National Institutes of General Medical Sciences (NIGMS). The content is solely the responsibility of the author(s) and does not necessarily reflect the views of the Burroughs Wellcome, Department of Energy, NIGMS, or the National Institutes of Health. The funders had no role in study design, data collection and analysis, decision to publish, or preparation of the manuscript.

**Competing interests:** I have read the journal's policy and the authors of this manuscript have the following competing interests: AS is a consultant and holds shares in a diagnostic start-up company, BioVector, Inc.

the activity of the CwlD amidase, which is found in all spore-forming bacteria. While *Bacillus subtilis* CwlD appears to have amidase activity on its own, *Clostridioides difficile* CwlD activity depends on its binding partner, the GerS lipoprotein. To understand why *C. difficile* CwlD requires GerS, we determined the X-ray crystal structure of the CwlD: GerS complex and discovered that GerS binds to a site distant from CwlD's active site. We also found that GerS stabilizes CwlD binding to its co-factor, $Zn^{2+}$, indicating that GerS allosterically activates CwlD amidase. Notably, regulation at the level of $Zn^{2+}$ binding has not previously been described for bacterial amidases, and GerS is the first protein to be shown to allosterically activate an amidase. Since binding partners of bacterial amidases were only first discovered 15 years ago, our results suggest that diverse mechanisms remain to be discovered for these critical enzymes.

## Introduction

Many spore-forming pathogens, such as *Clostridioides difficile*, rely on germination to initiate infection. *C. difficile* is a toxin-producing bacterium that is the leading cause of healthcare-associated infections in many developed countries [1,2]. As with many spore-forming bacteria, successful germination depends on the degradation of the thick protective layer of modified peptidoglycan known as the cortex. The cortex functions to maintain spore metabolic dormancy [3] by keeping spores in a partially dehydrated state [3], so its removal is critical for metabolism to resume during germination. Cortex degradation is mediated by cortex lytic enzymes that recognize a cortex-specific peptidoglycan modification known as muramic-δ-lactam (MAL). This strict substrate specificity ensures that cortex lytic enzymes degrade the cortex and not the germ cell wall [3,4].

MAL residues are generated once the cortex layer is synthesized during sporulation [5]. The sequential action of two conserved enzymes, the CwlD amidase and PdaA deacetylase, produce MAL. CwlD removes the peptide side chain from N-acetylmuramic acid (NAM) [6–8] so that PdaA can deacetylate NAM and catalyze lactam ring formation to form MAL [6,9,10]. Mutants that lack MAL due to loss of either of these cortex-modifying enzymes exhibit severe germination defects in both *Bacillus subtilis* and *C. difficile* [6,8–11]. In the case of *C. difficile*, MAL-deficient mutants also exhibit virulence defects in animal models of infection [9,12], highlighting the critical importance of this process.

While the functions of CwlD and PdaA are conserved between *B. subtilis* and *C. difficile* [6,8–10], we recently showed that *C. difficile* CwlD amidase activity depends on the GerS lipoprotein [6,12]. Notably, GerS is unique to the Peptostreptococcaceae family, suggesting that CwlD amidase activity in this family may be differentially regulated relative to other spore-forming organisms [6,12]. Indeed, *B. subtilis* CwlD does not appear to require additional partners for its function [7]. These observations raise the question as to why *C. difficile* CwlD function depends on GerS. Given that GerS and CwlD directly interact [6], GerS binding to CwlD presumably regulates CwlD activity.

CwlD is a member of the amidase_3 family (PF01520), which play critical roles during cell division in many bacteria (e.g AmiC) [13]. Amidase_3 family members are $Zn^{2+}$-dependent N-acetylmuramoyl amidases that use a $Zn^{2+}$ ion as part of their catalytic mechanism [14]. The activity of these amidases is tightly regulated in many bacteria because their aberrant cleavage of the peptidoglycan layer can result in cell lysis [15,16]. Studies in the past ~10 years have revealed novel mechanisms for regulating amidase activity, with binding partners appearing to allosterically regulate amidase function [17]. In many Gram-negative bacteria, the highly

conserved amidases, AmiA, AmiB, and AmiC, are activated upon binding regulators like EnvC, NlpD, and the recently discovered ActS [16,18–20]. These regulators likely displace an auto-inhibitory loop unique to this subset of amidases from the amidase active site [21,22]. In addition, the activity of the *Staphylococcus aureus* LytH amidase was recently shown to depend on the transmembrane protein, ActH [15]. While the mechanism by which ActH licenses LytH's ability to denude peptidoglycan remains unclear, LytH's ability to cleave stem peptides may be restricted to uncrosslinked peptidoglycan [15]. Taken together, these data suggest that amidase activation by binding partners may be a more widespread phenomenon that initially thought.

To gain insight into the molecular mechanism by which GerS regulates CwlD amidase activity, we solved the crystal structure of the CwlD:GerS complex. This is the first structure of an amidase bound to its regulator, and it reveals that GerS forms a homodimer that binds two CwlD monomers at a site opposite from CwlD's active site. Since *C. difficile* CwlD adopts a typical amidase_3 fold in the complex, GerS binding likely impacts CwlD amidase activity at a distance. Indeed, our biochemical analyses indicate that *C. difficile* CwlD does not bind $Zn^{2+}$ stably on its own in contrast with previously characterized amidase_3 family members. Instead, GerS binding to CwlD appears to promote CwlD binding to $Zn^{2+}$, suggesting a novel mechanism for allosterically regulating amidase_3 family member function. Taken together, our analyses expand our understanding of the diverse mechanisms used by bacteria to regulate the activity of peptidoglycan amidases via protein-protein interactions.

## Results

### Analyzing the dimerization of CwlD and GerS and the stoichiometry of the CwlD:GerS complex

To investigate why CwlD activity depends on GerS, we sought to biochemically characterize the CwlD:GerS complex and identify residues required to mediate this interaction using X-ray crystallography. Constructs encoding the soluble domains of CwlD and GerS were over-expressed in *Escherichia coli* and a co-affinity purification strategy was used to purify untagged GerS bound to His-tagged $CwlD_{\Delta 1-25}$ as previously described [6]. These domains correspond to CwlD and GerS lacking their predicted transmembrane (residues 1–25) and signal peptide/lipobox (residues 1–22) domains, respectively. The stability of this complex was analyzed using size exclusion chromatography (SEC) by comparing the elution profile of the complex relative to its individual CwlD and GerS proteins, which were both purified as His-tagged variants. $CwlD_{\Delta 25}$-His$_6$ alone primarily eluted as a monomer in solution, with a major peak corresponding to an apparent molecular weight (MW) of 33 kDa (**Fig 1A**) and a second less prominent peak corresponding to an apparent MW of 67 kDa, suggesting that CwlD can dimerize. Coomassie staining of the elution fractions confirmed that CwlD was the major protein detected in both peaks (**Fig 1A**). $GerS_{\Delta 22}$-His$_6$ alone eluted in 3 separate peaks, with apparent MWs of 23, 45 and 81 kDa (**Fig 1B**), strongly suggesting that GerS has the capacity to multimerize. The 23 kDa peak likely represents the monomeric form of the protein, since it corresponds to the expected MW; the 45 kDa peak likely corresponds to a dimer; and the 81 kDa peak likely represents a trimer or tetramer of GerS (**Fig 1B**). Coomassie staining of the three peaks confirmed that they contained predominantly GerS. Together, these results show that CwlD and GerS are mostly monomeric in solution but that GerS readily forms higher order complexes.

SEC analysis of the CwlD:GerS complex revealed that it primarily eluted as a complex with a calculated MW of 72 kDa (1$^{st}$ peak, **Fig 1C**). This result would be consistent with a ternary complex of a GerS dimer (46 kDa) with a CwlD monomer (28 kDa), or vice versa. However,

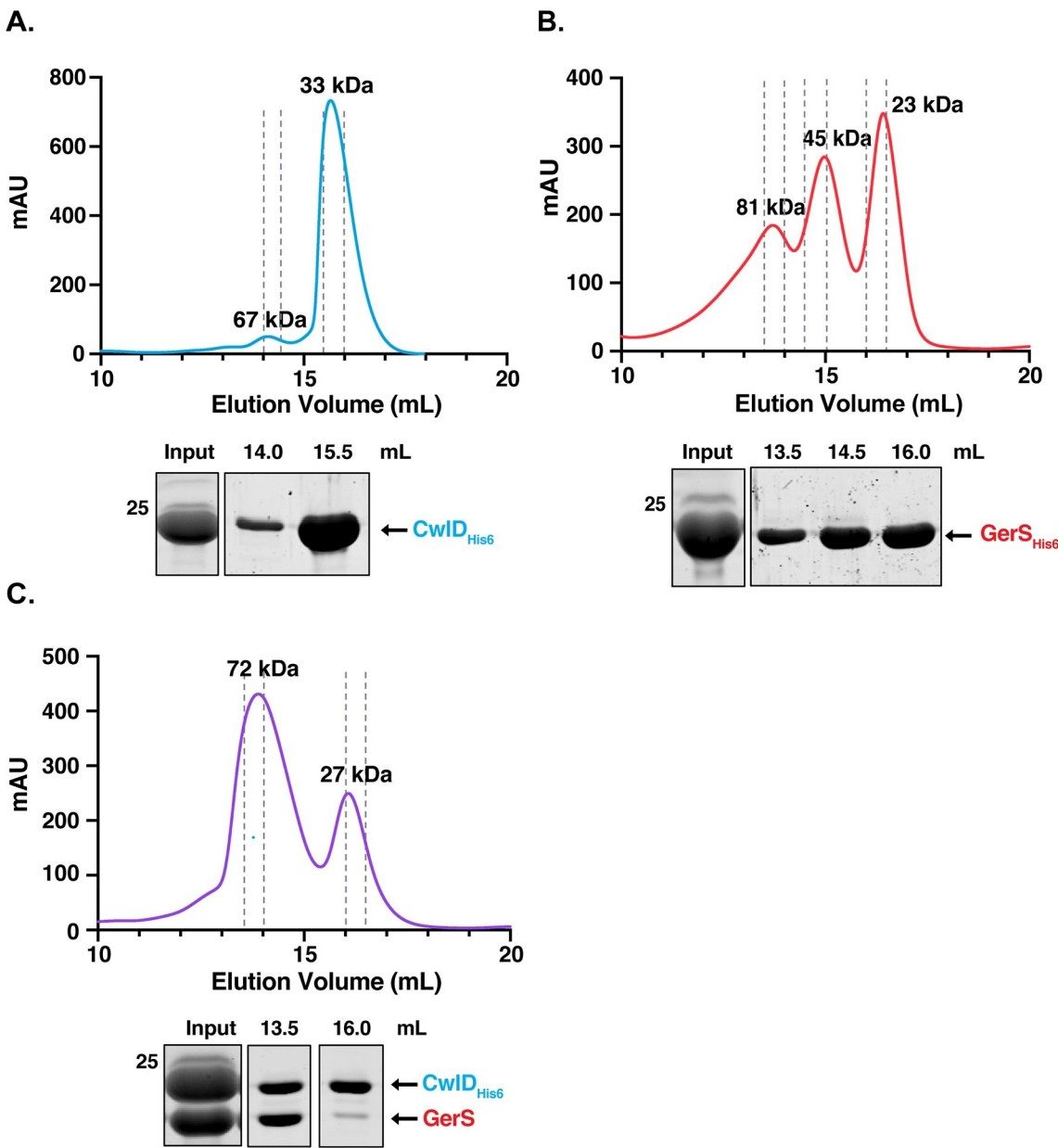

**Fig 1. Size Exclusion Chromatography analyses of CwlD, GerS, and the CwlD:GerS complex.** Purified $CwlD_{\Delta25}$-$His_6$ (A), $GerS_{\Delta22}$-$His_6$ (B), and the $CwlD_{\Delta25}$-$His_6$:$GerS_{\Delta22}$ complex (C) were analyzed by size exclusion chromatography. mAU corresponds to the UV absorbance measurements ($A_{280}$) during the protein elution. The estimated molecular weights of the protein species are indicated at the top of each peak. The input sample is derived from the co-affinity purification high imidazole elution fraction. This input sample as well as the fractions (0.5 mL) from the indicated elution volumes were analyzed by SDS-PAGE and Coomassie staining. Dashed grey lines indicate the fractions collected. The data shown are from one biological replicate, which is representative of three biological replicates performed independently. The molecular weight (MW) marker is shown on the left-hand side of the gel.

Coomassie staining of the SEC fractions revealed that the two proteins are present at a 1:1 ratio, suggesting that they bind stoichiometrically. In addition to this major peak, a smaller elution peak was observed that corresponds to an apparent MW of 27 kDa. Coomassie staining indicated that this peak primarily contained His-tagged $CwlD_{\Delta25}$ alone (**Fig 1C**). Since the input sample shows more CwlD than GerS, this band likely corresponds to free His-tagged $CwlD_{\Delta25}$ that is not bound to $GerS_{\Delta22}$.

### Crystal structure of the CwlD:GerS complex

**Overall complex structure.** To gain atomic-level insight into how GerS activates CwlD activity, we crystallized the SEC-purified CwlD:GerS complex and determined its structure. The asymmetric unit contained one molecule of CwlD and one monomer of GerS with the CwlD structure being typical of the amidase superfamily. GerS is largely anti-parallel beta sheet with the exception of an N-terminal alpha helix and forms a cupped hand-like shape similar to the LolA family of lipid-associated proteins [23]. While the members of the LolA family are typically monomeric [24], completion of the beta fold for GerS required two molecules such that the N-terminal alpha helix through the first 2 beta strands comes from one molecule in the crystal and the remainder of the GerS fold comes from a symmetry mate (**Fig 2A** and **2B**). Notably, both GerS molecules provide contacts to a single CwlD. This crossover-dependent assembly of the GerS fold is the same as was observed for the domain swapped assembly structure of the F47E mutant of LolA (PDBID 6FHM) [25]. Thus, the likely biological assembly for the CwlD:GerS complex contains two molecules of GerS and two CwlD and is in

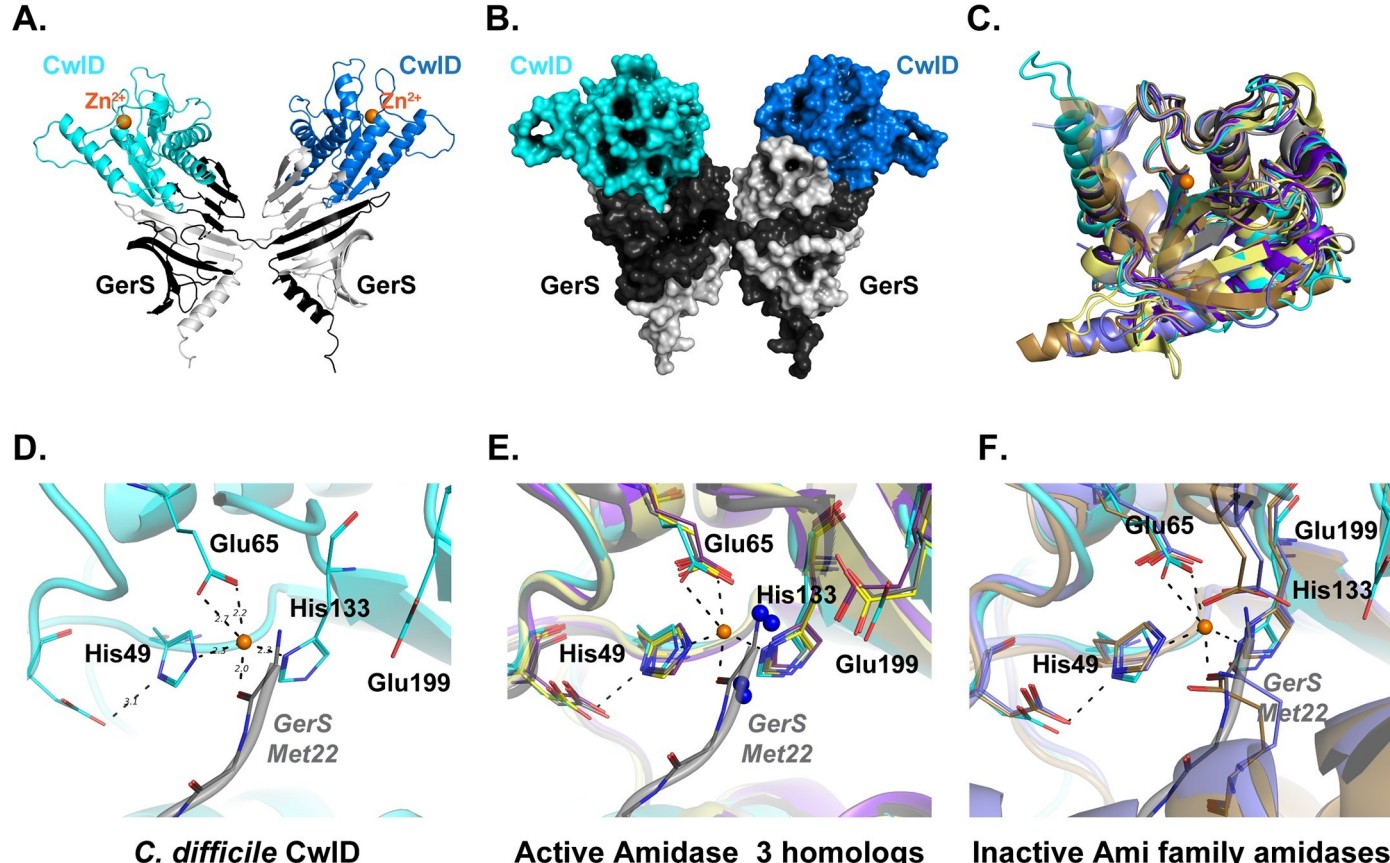

**Fig 2. Structure of the CwlD:GerS complex.** (A) Shown is the cartoon representation of the CwlD:GerS complex within the crystal with CwlD in blue tones and GerS in grey. The crystallographic asymmetric unit contains a single protomer of each with the 2 GerS/2 CwlD assembly formed by domain swapping of the GerS monomers. (B) The assembly is shown in space filling representation. (C) Least squares superposition of CwlD (cyan throughout) and its homologs performed using Superpose [44] in CCP4. (D) Close-up view of the metal binding site for CwlD with GerS from a fortuitous crystal packing (grey) providing coordination of the ion. (E) CwlD homolog structures (CwlD *Paenibacillus polymyxa* PDBID 1JWQ, unpublished; *C. difficile* CD2761 amidase_3 family member, 4RN7, unpublished; *Nostoc punctiforme*, AmiC2, 5EMI [29]; and *C. difficile* Cwp6, 5J72 [52]) in the active form have the $Zn^{2+}$ atom accessible to the solvent. They also have water molecules serving as 1–2 ligands (blue spheres), one of which is coordinated by a conserved glutamate. (F) CwlD homolog structures (*Bartonella henselae* AmiB PDBID 3NE8 [22] and *E. coli* AmiC 4BIN [21]) in the inactive forms have residues from their respective inhibitory helices providing $Zn^{2+}$ ligands.

agreement with both dynamic light scattering (DLS) data (**S1 Fig**) and size exclusion chromatography (**Fig 1**). The combined buried surface area upon binding of CwlD to the two GerS molecules is 1415 Å$^2$ and contains 10 hydrogen bonds and 6 salt bridges as calculated using PISA [26]. This interface is analogous to the interface observed for the C-terminal domain of the anti-anti-sigma factor MucB from *P. aeruginosa* (PDB ID 6IN8) [27] and the stress response factor RseB from *E. coli* (PDB ID 3M4W) [28].

Both CwlD and GerS have N-terminal sequences used for membrane association that were omitted for the crystallization constructs though neither would be occluded in the complex and would be free to extend to the membrane. For GerS, this would be an extension of the alpha helix on the periphery of the complex. For CwlD, the extension sits surface-exposed adjacent to the complex interface. Importantly, the binding orientation for CwlD relative to GerS leaves the CwlD Zn$^{2+}$-containing active site solvent accessible to substrate rather than at the complex interface.

**The CwlD structure.**    The structure of CwlD is the mixed alpha-beta fold expected for the N-acetylmuramoyl-L-alanine amidase superfamily (**Fig 2C**), with 3 conserved glutamate residues and 2 histidines coordinating the Zn$^{2+}$ ion (**Fig 2D**). His49 (positioned by Glu53), His133, and the carboxylate of Glu65 provide four of the ligands of the near octahedral ion coordination with distances of 2.2 Å. Zn$^{2+}$ coordination falls into two classes for the amidase family: (i) the active form where water molecules provide the 2 additional ion ligands, one of which coordinated by the conserved Glu199 position in CwlD (**Fig 2E**) and (ii) the autoinhibited form where a glutamate or glutamine from the inhibitory helix conserved in Ami family amidases [21,22] is liganded (**Fig 2F**). While CwlD does not have an autoinhibitory helix, the N-terminus of a GerS symmetry molecule provides the additional liganding for the Zn$^{2+}$. As this symmetry molecule is not one from the 2 GerS monomers used to bind CwlD, the interaction is not physiologically relevant. Instead, it is a fortuitous crystal packing event resulting in the displacement of the water molecules expected to coordinate the Zn$^{2+}$ in the active form of CwlD in solution.

**The GerS structure and CwlD interface.**    The heavy atom-phased solutions revealed that the functional folding of GerS for binding to CwlD requires two molecules of GerS. The LolA-like conserved anti-parallel beta sheet of GerS folds into a pseudo half beta barrel utilizing 11 strands where CwlD sits. The topology for the strands of the beta fold is strand numbers 6-5-4-3-2-1-11-10-9-8-7 where strands 1 and 2 at the core of the fold come from the second molecule of GerS. The interaction of the N-terminal alpha helix and the embedding of beta stands 1 and 2 into the symmetry mate generates an extensive buried surface area of 3766 Å$^2$ and is composed of 57 hydrogen bonds and 2 salt bridges involving residues 22 through 71 at which point crossover occurs. Alpha helix 1 (residues 65–82) and the C-terminal alpha helix (residues 217–235) of CwlD constitute the most significant interface with GerS contacting stands 2 through 6. A series of hydrophobic residues from strands 4–6 provide complementary packing for Trp233, Ile227, Tyr226 and Leu79 of CwlD. Numerous other interactions of interest occur at the CwlD:GerS interface. Arg169 from CwlD forms a salt bridge with Asp106 from strand 6 of GerS while His84 from the loop connecting strands 3 and 4 of GerS creates a communication network via Glu78 of helix 1 of CwlD to His61 of strand 2 of GerS.

**Zn$^{2+}$ stabilizes the CwlD:GerS complex.**    Our finding that CwlD in the co-crystal structure strongly resembles active amidase_3 family members raises the question as to whether GerS binding to CwlD induces conformational changes in CwlD and whether this binding event licenses CwlD activity. Our attempts to gain insight into this former question failed because we were unable to crystallize *C. difficile* CwlD$_{\Delta25}$-His$_6$ on its own (**Fig 1A**) even though CwlD$_{\Delta25}$-His$_6$ was stably folded in these DLS analyses (**S1 Fig**). Interestingly, DLS analyses of the SEC-purified CwlD:GerS complex revealed two peaks, one with a predicted radius

of 120 kDa (which would correspond to a dimer of dimers of CwlD:GerS) and one that appeared to be aggregated protein. This latter peak may represent GerS that has dissociated from the SEC-purified complex.

To identify factors that might affect the stability of GerS binding to CwlD, we tested the effect of disrupting (i) CwlD binding to $Zn^{2+}$ and (ii) CwlD's $Zn^{2+}$-dependent catalytic mechanism on the CwlD:GerS interaction. To disrupt CwlD's ability to bind $Zn^{2+}$, we generated constructs encoding alanine mutations of CwlD's $Zn^{2+}$-binding residues, His49 and Glu65. To disrupt CwlD's catalytic mechanism, we mutated the putative catalytic Glu199 residue to alanine. We then co-produced these His-tagged CwlD variants in *E. coli* and measured their ability to pull-down untagged GerS using the co-affinity purification assay.

CwlD-His$_6$ variants carrying the $Zn^{2+}$-binding mutations (H49A and E65A) reduced the amount of untagged GerS that co-purified, although it was still possible to purify the mutant CwlD:GerS complexes (**Fig 3A**). In contrast, wild-type levels of GerS co-purified with the E199A variant. When the co-affinity purifications were analyzed by SEC, the CwlD$_{H49A}$:GerS and CwlD$_{E65A}$:GerS complexes (**Fig 3B** and **3C**) eluted later than complexes containing either wild-type CwlD or CwlD$_{E199A}$ (**Fig 3D**). CwlD$_{H49A}$:GerS and CwlD$_{E65A}$:GerS complexes eluted with a calculated MW of 55 and 60 kDa, respectively, while the wild-type CwlD:GerS complex eluted with a predicted MW of 72 kDa (**Fig 3B** and **3C**). Since the elution volume of proteins by SEC depends on both their shape and MW, these observations suggest that the shape and/ or stoichiometry of CwlD:GerS complexes changes when CwlD is able to bind $Zn^{2+}$. However, Coomassie staining of the different fractions revealed that both proteins are present at ratios close to 1:1, irrespective of the CwlD point mutations (**Fig 3E**). Based on these observations, loss of $Zn^{2+}$-binding likely de-stabilizes the CwlD:GerS complex, although we cannot rule out the possibility that the CwlD $Zn^{2+}$ binding mutations affect the stoichiometry of CwlD binding to GerS.

To directly assess the effect of $Zn^{2+}$ on the stability of the CwlD:GerS complex, co-affinity purified CwlD:GerS complex was incubated for one hour in the presence of either a five-fold excess of $Zn^{2+}$ or 1 mM EDTA and then analyzed by SEC (**Fig 4A**). In the presence of $Zn^{2+}$, the wild-type CwlD:GerS complex primarily eluted at a higher predicted MW (83 kDa), while incubation with EDTA resulted in three peaks that corresponded to predicted MWs of 65, 45, and 28 kDa (**Fig 4A**). Although Coomassie staining indicated that the complex was still present in the 65 and 45 kDa peaks, GerS appeared to be present at higher levels in the 45 kDa peak than CwlD, suggesting that this peak may comprise GerS dimers along with CwlD alone. Since the 28 kDa peak was primarily comprised of His-tagged CwlD, these analyses strongly suggest that removing $Zn^{2+}$ from the CwlD:GerS complex via EDTA treatment promotes the dissociation of the complex. Notably, when the CwlD$_{H49A}$:GerS complex was incubated with $Zn^{2+}$ or EDTA, no changes in the SEC trace were observed (**Fig 4B**). Collectively, these results reinforce the notion that $Zn^{2+}$ coordination affects the stability of the CwlD:GerS complex.

To confirm that the CwlD $Zn^{2+}$-binding residue mutations disrupt binding to $Zn^{2+}$, we analyzed the $Zn^{2+}$ content of SEC-purified wild-type CwlD:GerS and CwlD$_{H49A}$:GerS complexes using inductively coupled plasma mass spectroscopy (ICP-MS). An average of about 500 ng $Zn^{2+}$/ mg protein was detected in the wild-type CwlD:GerS, while only background levels of $Zn^{2+}$ were detected in the CwlD$_{H49A}$:GerS complex (**Fig 5**). The same result was observed for a CwlD:GerS complex harboring an alanine mutation in the CwlD $Zn^{2+}$-binding residue, Glu65 (**S1 Table**). To validate the effects of these mutations on the individual proteins, we purified His-tagged wild-type CwlD and CwlD$_{H49A}$ on their own (**S3 Fig**), as well as His-tagged GerS alone (**Fig 1B**), and quantified the $Zn^{2+}$ levels in these individual proteins. Intriguingly, no $Zn^{2+}$ was detected in any of these samples, including with wild-type CwlD alone (**Fig 5**). This latter result was unexpected because $Zn^{2+}$ is present in the crystal structures

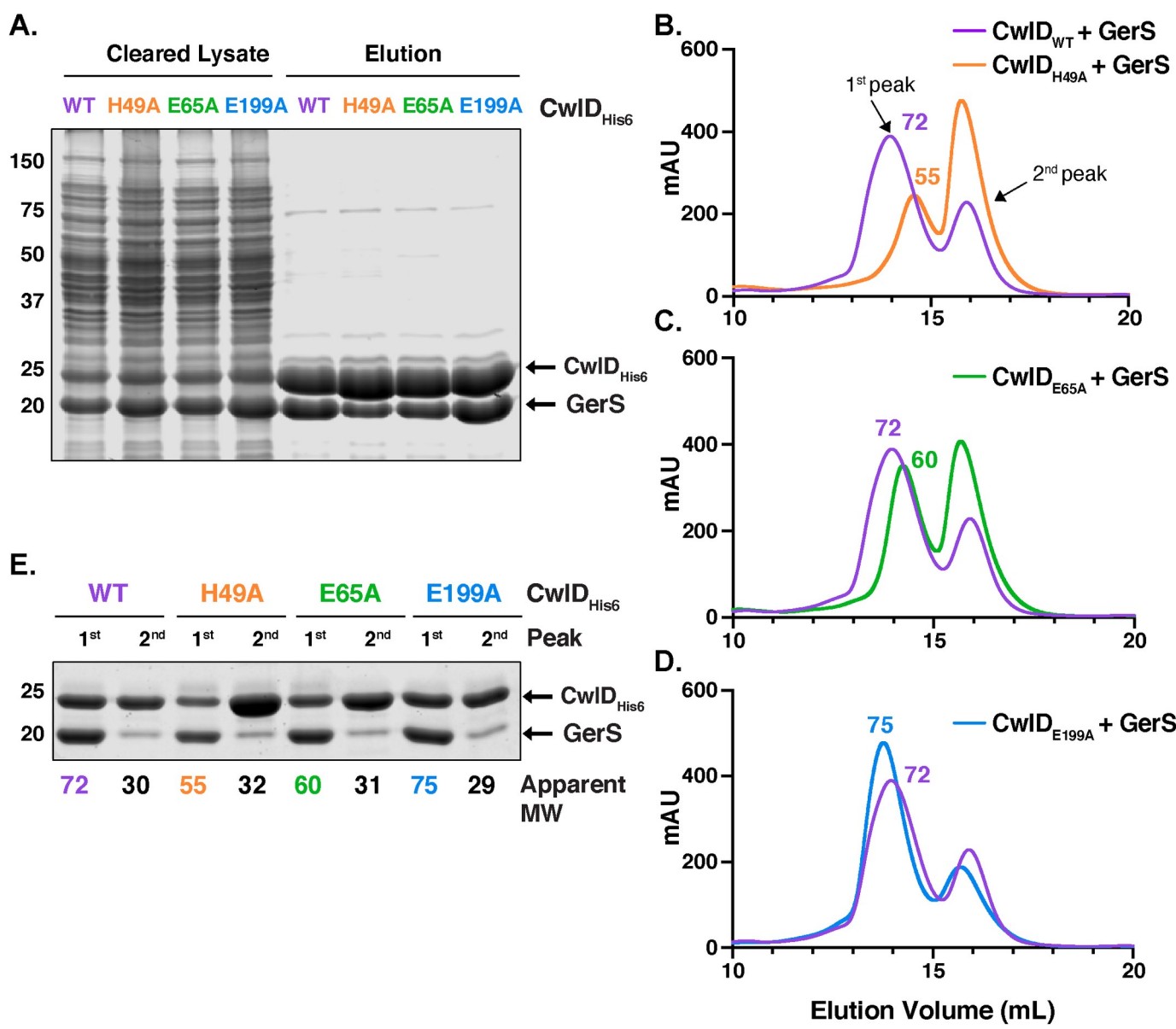

**Fig 3. Mutation of CwlD Zn$^{2+}$-binding residues, His49 and Glu65, decreases the stability of the CwlD:GerS complex.** (A) Coomassie stain of wild-type CwlD-His$_6$, H49A, E65A or E199A His-tagged CwlD variants co-affinity purified with untagged GerS. The indicated proteins were produced in *E. coli* and purified using Ni$^{2+}$- affinity resin. Cleared lysate and eluate fractions were analyzed using Coomassie staining. Comparison of the size exclusion chromatography traces of the wild-type CwlD:GerS complex with CwlD$_{H49A}$:GerS (B), CwlD$_{E65A}$:GerS (C) and CwlD$_{E199A}$:GerS (D) complexes. mAU corresponds to the UV absorbance measurements (A$_{280}$) during the protein elution. The 1$^{st}$ peak of each trace corresponds to the protein complex; the estimated molecular weight (MW) in kDa of the complex is given on top of this first peak. (E) Coomassie stain of fractions corresponding to the 1$^{st}$ and 2$^{nd}$ peak of each trace. The numbers at the bottom correspond to the apparent MW of each peak. The data shown are from one biological replicate, which is representative of three biological replicates performed independently.

of many amidase_3 family members even though exogenous Zn$^{2+}$ was not added to the expression or purification buffers in these studies [21,22,29]. Because of this surprising result, we purified another *C. difficile* amidase_3 homolog, CD2761, which binds Zn$^{2+}$ in X-ray crystallography analyses (PBD ID 4RN7, unpublished). Even more Zn$^{2+}$ was detected bound to CD2761 (~1100 ng Zn$^{2+}$/mg protein) than to WT CwlD:GerS (~450 ng Zn$^{2+}$/mg protein). Thus, if CwlD was able to bind Zn$^{2+}$ stably on its own, we would expect to detect it using this

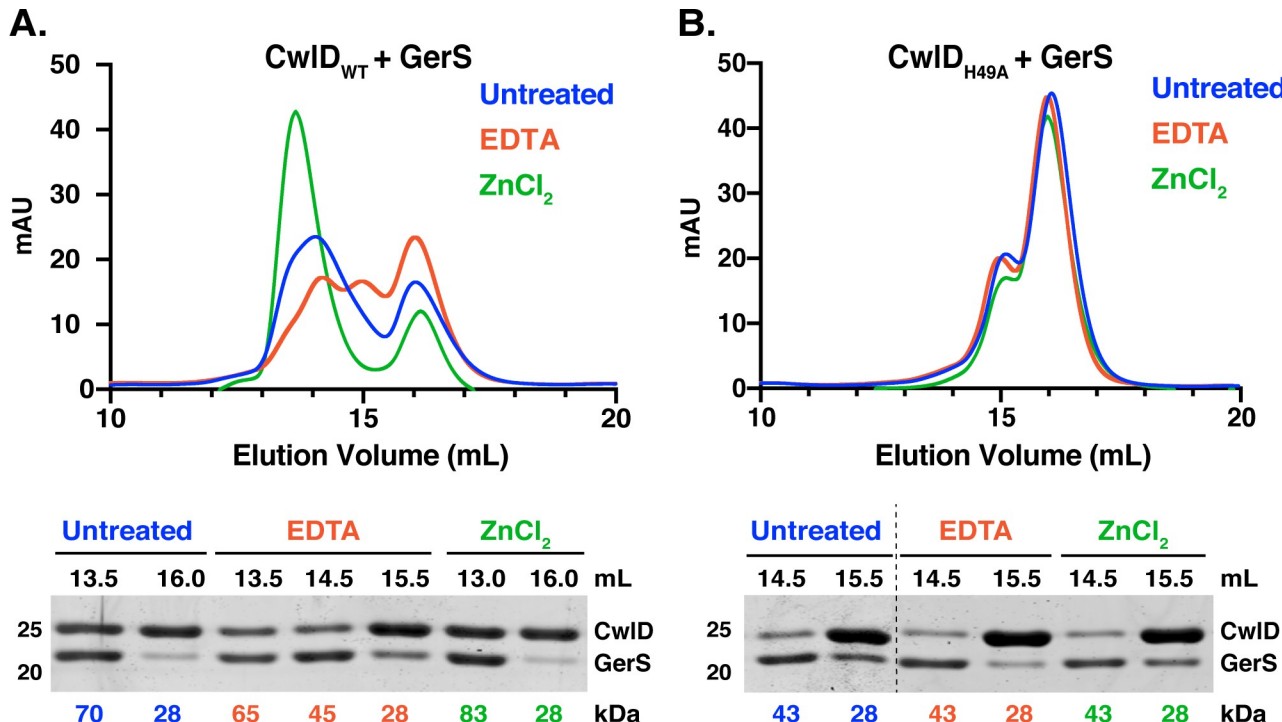

**Fig 4. Zn²⁺ stabilizes the CwlD:GerS complex.** Purified wild-type CwlD:GerS (A) and CwlD$_{H49A}$:GerS (B) complexes were analyzed by size exclusion chromatography after treatment with a five-fold excess of Zn²⁺Cl₂ (green traces) or 1 mM EDTA (red traces) for 1 hr. mAU corresponds to the UV absorbance measurements (A$_{280}$) during the protein elution. Fractions (0.5 mL) corresponding to the indicated elution volumes were analyzed by SDS-PAGE. The numbers at the bottom of the gels correspond to the apparent MW of each peak. The data shown are from one biological replicate, which is representative of three biological replicates performed independently.

method. Taken together, our results indicate that *C. difficile* CwlD behaves differently from all previously characterized amidase_3 family members to our knowledge in that it does not stably bind to Zn²⁺. Instead, GerS appears to promote CwlD binding to Zn²⁺. Conversely, CwlD binding to Zn²⁺ stabilizes the CwlD:GerS complex (compare CwlD$_{H49A}$ and CwlD$_{E65A}$ to CwlD$_{WT}$), **Figs 3** and **5**), suggesting a mutual dependence between CwlD, Zn²⁺, and GerS binding at least *in vitro*.

## CwlD binding to Zn²⁺ is required for amidase activity and stabilizes both CwlD and GerS in mature spores

We next wanted to confirm that the Zn²⁺-binding residues and the presumed catalytic glutamate 199 are required for CwlD function in *C. difficile*. To this end, we designed *cwlD* constructs encoding mutations in the Zn²⁺-coordinating residues, His49 and Glu65, and in the putative catalytic Glu199 residue. We then assessed the ability of these constructs to complement the germination defect of a *C. difficile ΔcwlD* strain. All complementation constructs used in this study were integrated into the ectopic *pyrE* locus using allele-coupled exchange [30] and expressed from their native promoters. To analyze germination, we used an optical density-based assay that measures the decrease in optical density of a population of germinating spores over time due to cortex hydrolysis and rehydration of the spore cytosol [31]. While the optical density of wild-type spores decreased by around 40%, the optical density of the *cwlD H49A, E65A* and *E199A* complementation strains did not change, similar to the parental *ΔcwlD* strain (**Fig 6A**). Taken together, these results indicate that His49, Glu65 and Glu199 are all required for CwlD function.

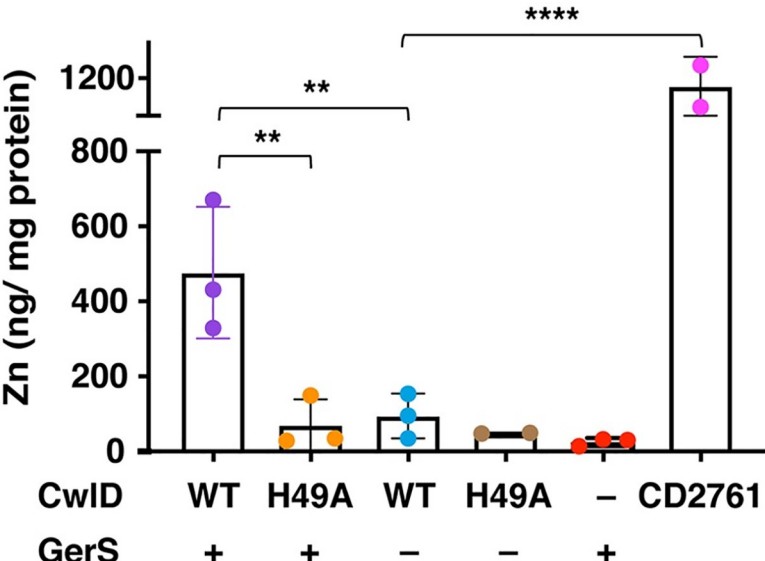

**Fig 5. Elemental analyses of CwlD and CwlD:GerS complex.** Elemental analyses of $Zn^{2+}$ in CwlD:GerS complexes corresponding to His-tagged wild-type CwlD:GerS and $CwlD_{H49A}$:GerS, and individual His-tagged proteins (wild-type CwlD, GerS and CD2761 and $CwlD_{H49A}$). Elemental analyses were performed on affinity-purified proteins produced in *E. coli* in the absence of $Zn^{2+}$ supplementation that were further purified using size exclusion chromatography (also in the absence of $Zn^{2+}$ supplementation). Each dot represents data from a given biological replicate. Three independent protein purifications were analyzed with the exception for $CwlD_{H49A}$ and CD2761, for which two independent protein purifications were performed. Bars represent the mean of the three (or two) independent experiments. Standard deviation is shown. Statistical significance was determined using a one-way ANOVA and Tukey's test. **, $p < 0.005$; ****, $p < 0.0001$.

Interestingly, the $Zn^{2+}$-binding mutations, H49A and E65A, decreased both CwlD and GerS levels in spores, while disruption of the putative catalytic glutamate E199A did not (**Fig 6C**). Notably, the $CwlD_{H49A}$ and $CwlD_{E65A}$ variants were detected at wild-type levels in sporulating cells, indicating that these $Zn^{2+}$-binding mutations do not affect the production or stability of CwlD (**Fig 6B**). Instead, they appear to affect CwlD incorporation or stability in mature spores. This result is consistent with our previous observation that loss of CwlD decreases GerS levels in mature spores, whereas loss of GerS does not affect CwlD levels [6]. Since the $Zn^{2+}$-binding mutations in CwlD did not affect the stability of recombinant His-tagged CwlD in SEC analyses (**S3 Fig**), our findings in *C. difficile* reinforce our biochemical data indicating that CwlD binding to $Zn^{2+}$ stabilizes CwlD:GerS complex formation, which apparently affects CwlD stability or incorporation in mature spores.

## Identification of a salt bridge critical for GerS binding to CwlD

To directly test whether GerS binding to CwlD stabilizes CwlD binding to $Zn^{2+}$, we sought to identify mutations that disrupt CwlD:GerS complex formation separate from the $Zn^{2+}$-binding mutations. Guided by the crystal structure of the complex, we identified two potential salt bridges that might promote CwlD:GerS binding: (i) CwlD arginine 169 (R169)-GerS aspartic acid 106 (D106) and (ii) CwlD glutamate 78 (E78)-GerS histidine 61 (H61) (**Fig 7A**). To assess the importance of these salt bridges in mediating binding between CwlD and GerS, we generated constructs encoding His-tagged CwlD variants carrying the following substitutions: arginine 169 to aspartic acid (R169D) or glutamine (R169Q) and glutamate 78 to glutamine (E78Q). We also generated untagged expression constructs encoding cognate salt bridge mutations in GerS, namely aspartate 106 to arginine (D106R) or alanine (D106A) and histidine 61

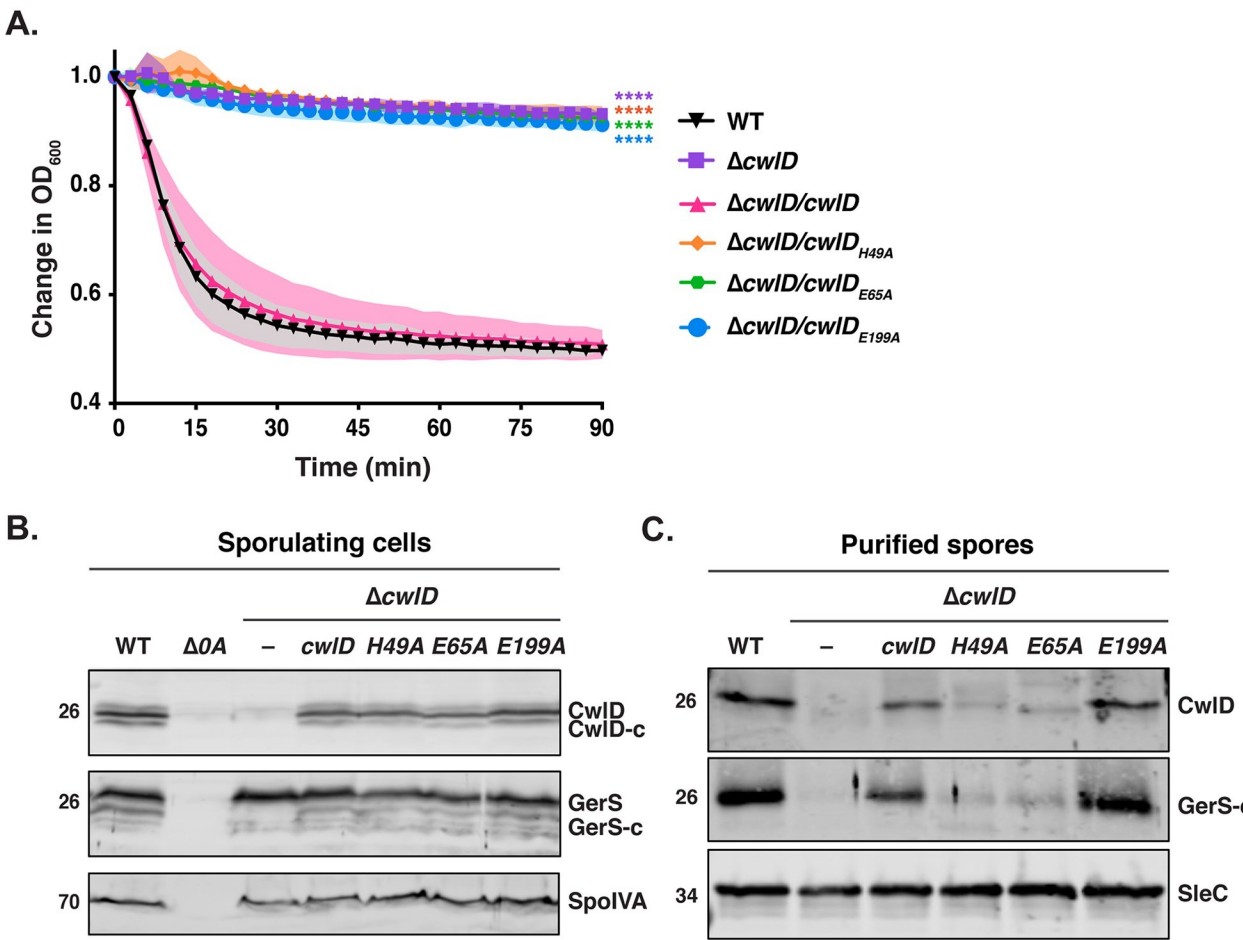

**Fig 6. Mutations in Zn²⁺ binding residues disrupt CwlD function and affect the stability of the CwlD:GerS complex in the spore.** (A) Spore germination as measured by the change in the $OD_{600}$ in response to germinant in *cwlD* mutants spores relative to wild-type spores. Purified spores were resuspended in BHIS, and germination was induced by adding taurocholate (1% final concentration). The ratio of the $OD_{600}$ of each strain at a given time point relative to the $OD_{600}$ at time zero is plotted. The mean $OD_{600}$ determined from three biological replicates from three independent spore preparations are shown. Shading represents the standard deviation as the area between error bars for each time point measured. Statistical significance relative to wild-type was determined using two-way ANOVA and Tukey's test. **** $p < 0.0001$. (B) Western blot analysis of CwlD and GerS levels in sporulating cells from wild type (WT), *ΔcwlD*, and *ΔcwlD* strains complemented with constructs encoding the CwlD variants H49A, E65A and E199A. His49 and Glu65 directly bind Zn²⁺ in the structure, while Glu199 is part of the amidase catalytic mechanism. CwlD-c and GerS-c indicate the cleaved form of both proteins. *Δspo0A* (*Δ0A*) was used as a negative control for sporulating cells, and SpoIVA was used as a loading control. (C) Western blot analysis of CwlD and GerS levels in spores from the strains used in the experiments described in (A). SleC was used as a loading control.

to alanine (H61A). We first co-produced the His-tagged CwlD variants in *E. coli* and measured their ability to pull-down untagged wild-type GerS using the co-affinity purification assay. CwlD variants carrying substitutions in Arg169 reduced GerS co-purification by 90% (**Figs 7B, 7C and S4**). Similarly, GerS variants carrying substitutions in Asp106 reduced GerS co-purifications with CwlD by ~90% also. Swapping the charges on the CwlD Arg169: GerS Asp106 interaction by combining the CwlD R169D and GerS D106R substitutions partially restored CwlD:GerS complex formation (**Fig 7B** and **7C**), further highlighting the importance of this salt bridge in mediating complex formation. In contrast, substitutions in residues comprising the second salt bridge did not strongly affect CwlD or GerS binding, since mutation of the CwlD$_{E78}$ or GerS$_{H61}$ did not affect the levels of GerS that co-affinity purified with His-tagged CwlD (**Fig 7D**). Taken together, these results implicate the CwlD$_{R169}$:GerS$_{D106}$ salt

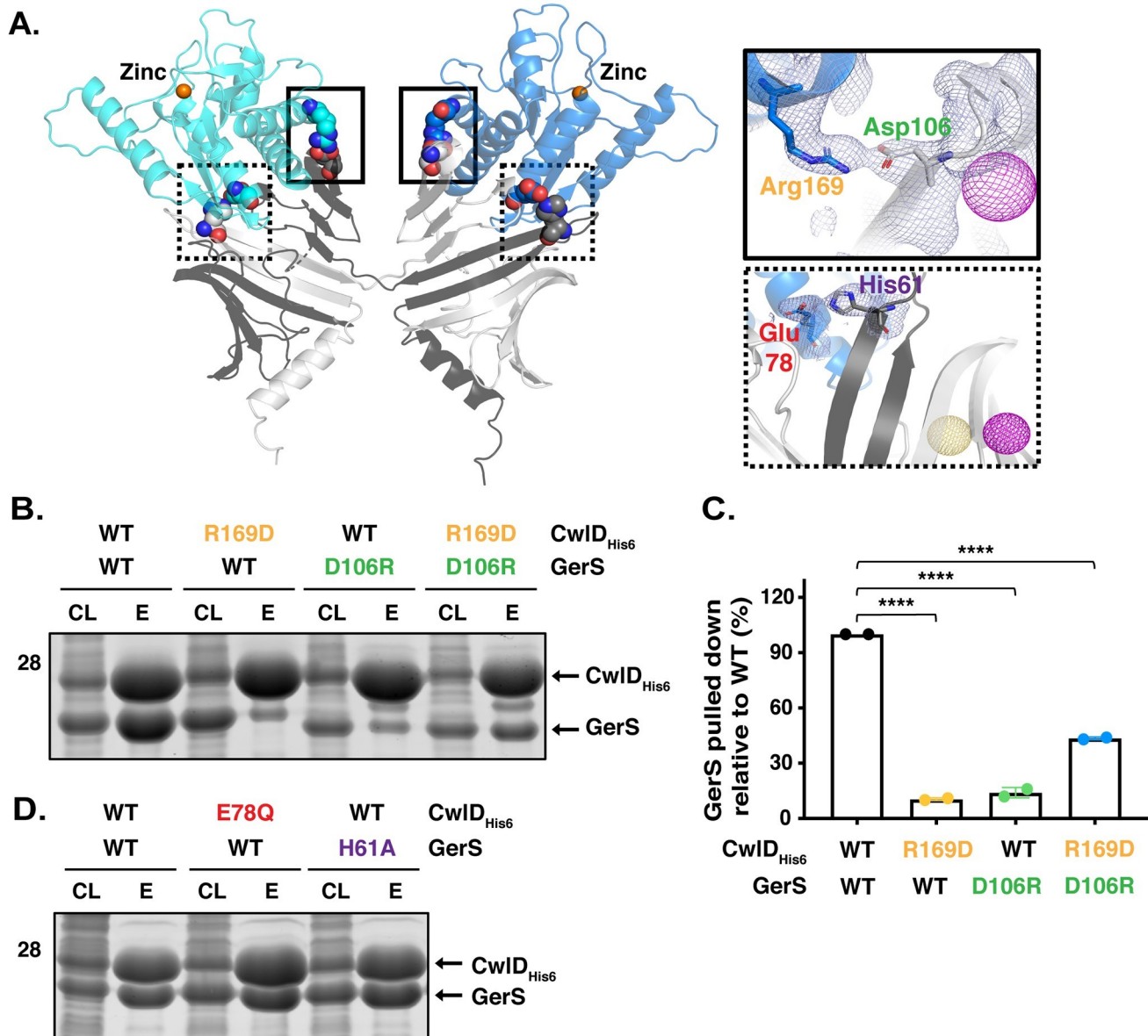

**Fig 7. A salt bridge between CwlD_{R169} and GerS_{D106} is important for CwlD:GerS binding.** (A) Specific interactions between CwlD and GerS observed in the structure. Salt bridge formation between CwlD Arg169 (R169) and GerS Asp106 (D106) is highlighted in the solid boxes and between CwlD Glu78 (E78) and GerS His61 (H61) in the dotted boxes. The light blue mesh shows the heavy atom phased electron density after density modification and building contoured at 1 sigma. The yellow mesh represents the anomalous signal at 3 sigma for the selenomethionine derivative and the maroon represents the sodium iodide derivative after scaling, outlier rejection and phasing within Autosharp (Global Phasing Limited). (B) Coomassie stain of co-affinity purifications of wild-type or R169D CwlD-His_6, with wild-type GerS or the D106R variant. The indicated proteins were produced in *E. coli* and purified using Ni^{2+}- affinity resin. Cleared lysate (CL) and eluate (E) fractions were analyzed using Coomassie staining. (C) Percentage of GerS pulled down with CwlD relative to the amount detected in the wild-type CwlD:GerS complex, which was set at 100%. The mean of two assays from two independent co-affinity purifications is shown, and dots represent each of those co-affinity purifications. Statistical significance relative to the wild-type complex was determined using a one-way ANOVA and Tukey's test. **** p < 0.0001. (D) Coomassie stain of cleared lysate and elution fractions from co-affinity purifications of His-tagged CwlD:GerS, CwlD_{E78Q}:GerS, or CwlD:GerS_{H61A}.

bridge in stabilizing the interaction between recombinant CwlD amidase and its binding partner, GerS.

To confirm the importance of the salt bridge in *C. difficile*, we analyzed CwlD:GerS binding using co-immunoprecipitation analyses that employed FLAG-tagged CwlD as bait. To this

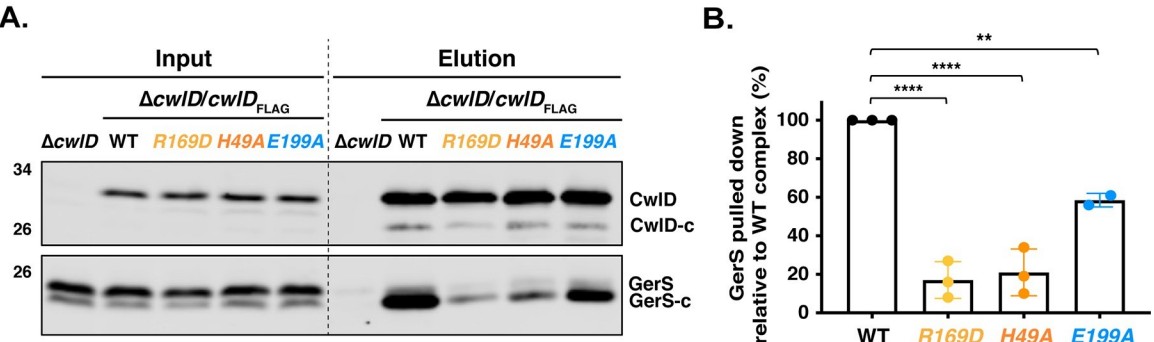

**Fig 8. Validation of the CwlD:GerS salt bridge by co-immunoprecipitation.** (A) Western blot of GerS in co-immunoprecipitation fractions obtained from pulling down FLAG-tagged proteins produced in sporulating cells for wild-type CwlD, $CwlD_{R169D}$, $CwlD_{H49A}$, or $CwlD_{E199A}$. CwlD was detected using an anti-FLAG antibody, while untagged GerS was detected using an anti-GerS antibody. CwlD-c and GerS-c indicate cleaved form of both proteins. (B) Percentage of GerS pulled down with FLAG-tagged CwlD relative to the amount detected in pull-downs with FLAG-tagged wild-type CwlD. A mean of three independent assays is shown, except for $CwlD_{E199A}$-FLAG, for which two independent biological replicates were performed. Dots represent the % of GerS pulled-down in each biological replicate. Statistical significance relative to the wild-type complex was determined using one-way ANOVA and Tukey's test. $^{**}$, $p < 0.005$; $^{****}$ $p < 0.0001$.

end, we generated complementation constructs encoding either wild-type or R169D FLAG-tagged CwlD ($CwlD_{FLAG}$). We then measured their ability to bind untagged GerS in sporulating *C. difficile* lysates. We also tested the effect of $Zn^{2+}$-binding (H49A) and catalytic residue (E199A) mutations on $CwlD_{FLAG}$:GerS complex formation by generating complementation constructs encoding FLAG-tagged variants carrying these mutations. When wild-type FLAG-tagged CwlD was immunoprecipitated from sporulating cell lysates, untagged GerS readily co-purified. In contrast, immunoprecipitation of $CwlD_{R169D}$ and $CwlD_{H49A}$ resulted in a 5-fold reduction in the amount of GerS pulled-down via FLAG-tagged CwlD (**Fig 8A** and **8B**), consistent with the co-affinity purification analyses in *E. coli* (**Fig 7B**). $CwlD_{E199A}$ retained the ability to bind GerS, although this interaction decreased ~40% relative to the wild-type CwlD:GerS complex (**Fig 8A** and **8B**). This finding could suggest that decreasing $Zn^{2+}$ coordination by a water molecule in the active site decreases the affinity of $CwlD_{E199A}$ for GerS in sporulating cells.

Interestingly, the primary form of GerS detected in co-immunoprecipitations with FLAG-tagged CwlD was the cleaved form of GerS (**Fig 8A**). As a lipoprotein, GerS is secreted across the outer membrane by its N-terminal signal peptide in sporulating cells [12]. Following secretion, GerS gets lipidated and then its signal peptide is removed through the action of the lipoprotein signal peptidase, Lsp [32]. Notably, while full-length GerS and cleaved GerS are detectable in sporulating cells, only cleaved GerS is detected in mature spores [12] consistent with its maturation as a lipoprotein [32]. These observations strongly suggest that GerS and CwlD interact *after* they are transported across the outer membrane. Altogether, our data indicate that stable CwlD:GerS complex formation depends on a salt bridge formed between $CwlD_{R169}$ and $GerS_{D106}$ and that stable binding between CwlD and GerS likely happens in the outer forespore membrane.

## Functional analyses of the salt bridge in *C. difficile*

To test whether the CwlD:GerS interaction is critical for the CwlD function in *C. difficile*, we analyzed the germination phenotype of the *cwlD* complementation strain producing FLAG-tagged $CwlD_{R169D}$. This strain was used in the co-immunoprecipitation assays described above (**Fig 8A**). We also assessed the effect of a GerS D106R substitution on spore germination by

complementing $\Delta gerS$ with a construct encoding FLAG-tagged $GerS_{D106R}$. The germination phenotypes of the $\Delta cwlD/cwlD_{R169D\text{-}FLAG}$ and $\Delta gerS/gerS_{D106R\text{-}FLAG}$ strains were analyzed using the optical density-based spore germination assay. While the optical density of the parental $\Delta cwlD$ and $\Delta gerS$ spores did not significantly decrease over the course of 90 min [6], the optical density of the $\Delta cwlD/cwlD_{R169D\text{-}FLAG}$ and $\Delta gerS/gerS_{D106R\text{-}FLAG}$ spores decreased by >40% over this time. $\Delta cwlD/cwlD_{R169D\text{-}FLAG}$ mutant spores germinated like wild-type spores and the wild-type complementation spores, $\Delta cwlD/cwlD_{FLAG}$ and $\Delta gerS/gerS_{FLAG}$ spores (**Figs 9A** and **S5A**), whereas $\Delta gerS/gerS_{D106R\text{-}FLAG}$ mutant spores germinated more slowly than wild-type and wild-type complementation spores (p < 0.001, **Fig 9A**).

To better understand the slower germination observed in this optical density-based assay for $\Delta gerS/gerS_{D106R\text{-}FLAG}$ spores, we visually monitored their germination using phase-contrast microscopy. One hour after inducing germination, all wild-type and $\Delta gerS/gerS_{FLAG}$ spores lost their phase-bright appearance (**S6A Fig**) as the cortex is degraded and the core hydrates [33]. In contrast, $\Delta gerS/gerS_{D106RFLAG}$ spores germinated more heterogeneously, with ~30% remaining phase-bright during this time period (**S6A** and **S6B Fig**). $\Delta gerS$ spores remained phase-bright during this time consistent with their inability to degrade the cortex, although some spore germination was detectable at the 90' timepoint [12].

We next analyzed the germination efficiency of the $cwlD$ and $gerS$ point mutants by plating spores on media containing germinant. While the $\Delta cwlD$ and $\Delta gerS$ parental strains exhibited a ~2 log decrease in spore germination efficiency relative to the wild-type strain (**Fig 9B**), no statistically significant defect in spore germination was observed for $cwlD_{R169D\text{-}FLAG}$ and $gerS_{D106R\text{-}FLAG}$ mutant spores relative to wild-type or the $\Delta cwlD/cwlD_{FLAG}$ and $\Delta gerS/gerS_{FLAG}$ complementation strain spores (**Fig 9B**).

Since CwlD enhances the levels of GerS in mature spores [6], and CwlD $Zn^{2+}$-binding mutations affect both CwlD and GerS levels in mature spores (**Fig 6**), we analyzed the levels of CwlD and GerS in the $cwlD_{R169D\text{-}FLAG}$ and $gerS_{D106R\text{-}FLAG}$ mutant spores. Interestingly, both the $cwlD_{R169D\text{-}FLAG}$ and $gerS_{D106R\text{-}FLAG}$ mutations led to lower levels of GerS in spores, while CwlD levels were unaffected by these mutations (**Fig 9D**). The $CwlD_{R169D\text{-}FLAG}$ variant also ran at a higher apparent MW than wild-type $CwlD_{FLAG}$. To ensure that the point mutations did not affect the stability of these proteins in sporulating cells, we analyzed the levels of these proteins in the different complementation strains using western blotting. $CwlD_{R169D\text{-}FLAG}$ levels were similar to those detected in the wild-type $\Delta cwlD/cwlD_{FLAG}$ complementation strain. Even though lower levels of $GerS_{D106R\text{-}FLAG}$ were detected using the GerS antibody in sporulating cells, no differences were found at the level of this protein with the anti-FLAG antibody (**Fig 9C**). This discrepancy suggests that our anti-GerS antibody binds the $GerS_{D106R}$ variant less efficiently than wild-type GerS. Consistent with the $GerS_{D106R}$ substitution causing a conformational change in GerS, $GerS_{D106R\text{-}FLAG}$ ran at a different apparent MW than wild-type $GerS_{FLAG}$ in sporulating cells. Taken together, these results indicate that disruption of the CwlD:GerS salt bridge decreases the stability of CwlD binding to GerS in both *E. coli* and *C. difficile* (**Figs 7** and **8**). In the case of the $GerS_{D106R}$ substitution, decreasing GerS binding to CwlD reduces CwlD function partially in optical density and phase-contrast-based germination assays (**Figs 9A** and **S6**). However, since the salt bridge mutations did not markedly impair spore germination despite severely decreasing CwlD:GerS binding, our results suggest that low levels of GerS are sufficient to license CwlD activity.

## Discussion

The activity of cell wall hydrolases like amidases must be tightly regulated during bacterial growth to avoid perturbations that can lead to cell lysis [34]. A wide array of mechanisms

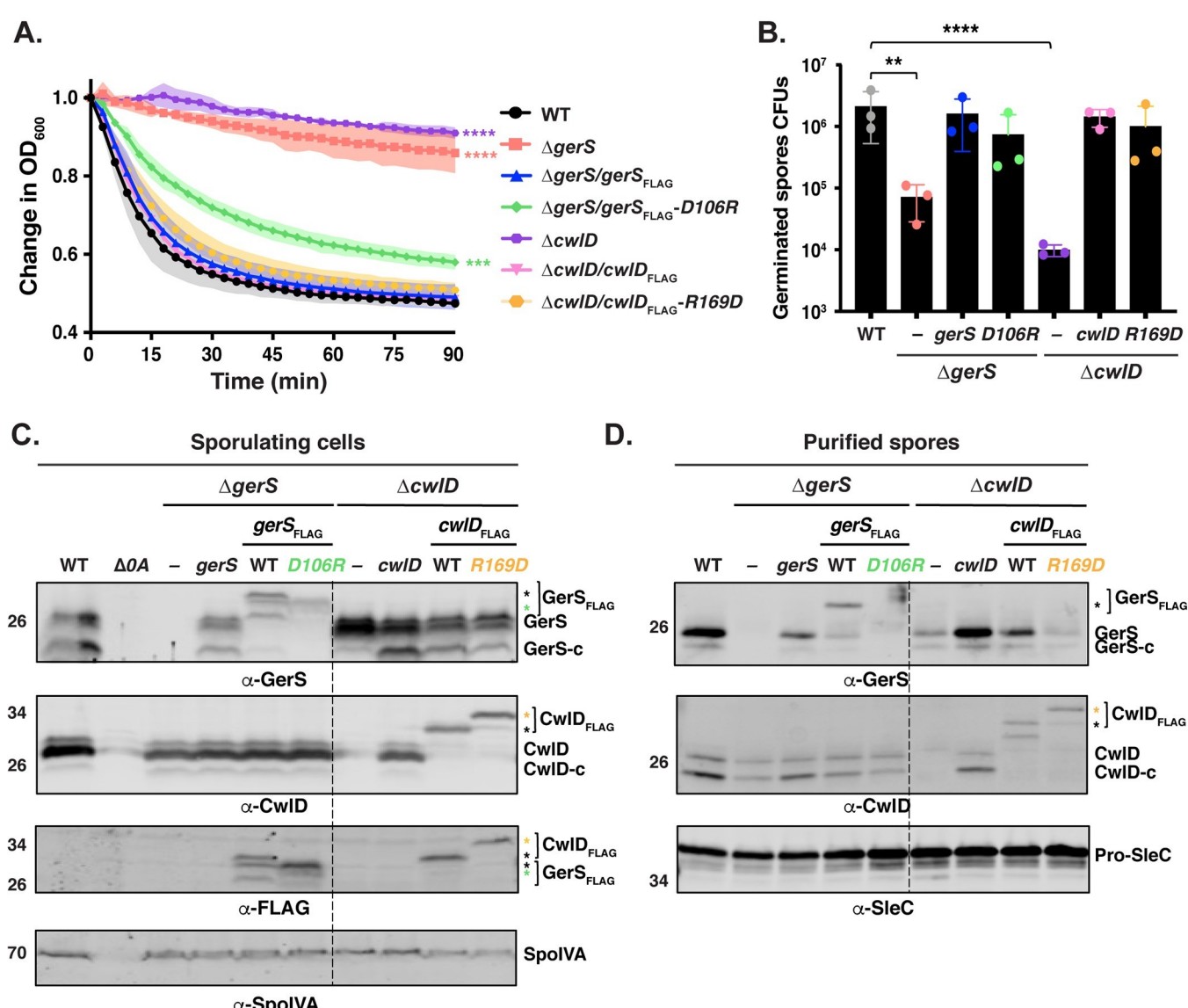

**Fig 9. A GerS_D106R mutation affects the stability of GerS protein and leads to slower germination.** (A) Spore germination as measured by the change in OD_600 in response to germinant in $gerS_{D106R}$ and $cwlD_{R169D}$ mutant spores relative to wild-type spores. Purified spores were resuspended in BHIS, and germination was induced by adding taurocholate (1% final concentration). The ratio of the OD_600 of each strain at a given time point relative to the OD_600 at time zero is plotted. The mean of three assays from 3 independent spore preparations are shown. Shading represents the standard deviation as the area between error bars for each time point measured. Statistical significance relative to wild-type was determined using two-way ANOVA and Tukey's test. ***, p< 0.0005; **** p < 0.0001. (B) Germination efficiency of $gerS_{D106R}$ and $cwlD_{R169D}$ point mutant spores. The number of colony forming units (CFUs) produced by germinating spores is shown. The mean OD_600 determined from three independent spore preparations is shown along with the associated standard deviations. Statistical analyses relative to the wild-type were performed using one-way ANOVA and Tukey's test. **, p < 0.005; ****, p < 0.0001. (C) Western blot analysis of CwlD and GerS levels in sporulating cells from the wild-type (WT), *ΔgerS*, *ΔgerS* complemented with either *gerS*, or *gerS* encoding either wild-type or D106R FLAG-tagged GerS variants as well as *ΔcwlD*, *ΔcwlD* complemented with either *cwlD* encoding either wild-type or R169D CwlD variants. CwlD-c and GerS-c indicate cleaved form of both wild-type proteins; asterisks indicate the cleaved form of FLAG-tagged. *Δspo0A* (*Δ0A*) was used as negative control for sporulating cells, and SpoIVA was used as a loading control. GerS and CwlD levels were analyzed using GerS, CwlD and FLAG antibodies. (B) Western blot analysis of CwlD and GerS levels in spores from the strains used in the experiments described in (C). SleC was used as a loading control.

control cell wall hydrolase activity, including transcription, protein localization, stability, peptidoglycan modification, and protein-protein interactions [17]. The regulation of cell wall amidases by protein-protein interactions was first reported only 12 years ago and was previously thought to be largely restricted to the AmiA/B/C amidases of Gram-negative bacteria [16,22]

until the activities of the LytH amidase of *Staphylococcus aureus* [15] and CwlD amidase of *C. difficile* (6) were found to depend upon binding partners. However, in the absence of structures of the amidases bound to their protein regulators, it remained unclear whether these amidases were allosterically regulated by their binding partners.

In determining the structure of the *C. difficile* CwlD amidase bound to its protein regulator, the GerS lipoprotein, and demonstrating that GerS binds CwlD at a site distant from CwlD's active site (**Fig 2**), we show in this study that the *C. difficile* CwlD amidase is allosterically activated by the Peptostreptococcaceae-specific GerS lipoprotein. Our biochemical data strongly suggest that GerS binding to CwlD promotes CwlD binding to $Zn^{2+}$, which directly participates in the catalytic mechanism of amidase_3 family members. Unlike all previously characterized Amidase_3 family members studied to our knowledge [15,17,21,22,29], SEC-purified CwlD does not bind $Zn^{2+}$ stably on its own in ICP-MS analyses, whereas $Zn^{2+}$ is readily detected in the SEC-purified CwlD:GerS complex (**Fig 5**). In addition, the stability of the CwlD:GerS complex depends on CwlD binding to $Zn^{2+}$ in both *E. coli* and *C. difficile* (**Figs 3, 4** and **6**), and disrupting CwlD $Zn^{2+}$ binding appears to destabilize CwlD in mature *C. difficile* spores (**Fig 5**). Thus, $Zn^{2+}$ plays a structural role in regulating CwlD activity beyond its direct role in catalysis. Consistent with this hypothesis, mutation of a salt bridge at the CwlD:GerS interface ($CwlD_{R169}$:$GerS_{D106}$) destabilizes the complex in *E. coli* and *C. difficile* (**Figs 7 and 8**), reduces GerS levels in mature spores, and impairs (albeit subtly) CwlD function in germination analyses of $\Delta gerS/gerS_{D106R}$ spores (**Figs 9** and **S6**). Notably, this salt bridge is conserved across Peptostreptococcaceae family members, suggesting that this interaction plays a similar role in related GerS homologs.

The salt bridge mutations provided further insight into when and where CwlD and GerS interact, since these mutations greatly reduced GerS levels in mature spores without affecting GerS levels in sporulating cells (**Fig 9**). Given that the co-immunoprecipitation data revealed that only the secreted, cleaved form of GerS binds full-length, FLAG-tagged CwlD (**Fig 8**), CwlD:GerS complex formation apparently stabilizes GerS in mature spores. Taken together, these results imply that the allosteric conduit connecting GerS to CwlD's active site functions as a two-way street, with CwlD binding to GerS affecting GerS's conformational stability, and GerS binding to CwlD controlling CwlD's ability to bind $Zn^{2+}$.

Based on these observations, we propose the following model. GerS binds CwlD only after GerS is secreted across mother cell-derived membranes and completes its lipoprotein maturation process. CwlD is also transported across mother cell-derived membranes via its predicted N-terminal transmembrane domain [35], but only its full-length, membrane-bound form binds lipidated GerS. This binding event likely triggers a conformational change in CwlD that promotes $Zn^{2+}$ binding, since $Zn^{2+}$ is only found when CwlD is complexed with GerS (**Fig 5**). This interaction may also help GerS orient CwlD's active site towards its cortex peptidoglycan substrate close to the membrane, since only the membrane-bound forms of these proteins interact.

Despite these conclusions, our study raises several questions regarding the regulation of CwlD amidase activity. Does GerS regulate CwlD amidase activity outside of GerS's effect on CwlD binding to $Zn^{2+}$? Does GerS need to remain bound to CwlD for CwlD to function as an amidase? Why does *C. difficile* CwlD require a binding partner when the *B. subtilis* CwlD can remodel peptidoglycan on its own when heterologously produced in *E. coli* [7]? Testing whether *B. subtilis* CwlD alone can bind $Zn^{2+}$ and whether the structures of these two CwlD homologs differ will provide important insight into these questions, although it is also possible that *B. subtilis* (or other organisms lacking GerS homologs) encode an unknown regulator that modulates CwlD activity. Finally, what is the signaling conduit that allows GerS binding to an allosteric site on CwlD to stabilize CwlD active site binding to $Zn^{2+}$?

Addressing this latter question would be greatly facilitated by structural characterization of CwlD alone. While we were unable to crystallize only CwlD, crystallographic analyses of Ami family amidases prior to their activation by protein regulators indicate that *E. coli* AmiC and *Bartonella henselae* AmiB are auto-inhibited by a regulatory α-helix that blocks access to their active sites [21,22]. Notably, CwlD homologs lack this regulatory helix (**Fig 2**), and amidases that lack this helix, such as *Nostoc punctiforme* AmiC2 and *Bacillus polymyxa* CwlV are active on their own [29,36]. While the precise mechanism by which the binding partners of Ami family amidases, EnvC, NlpD, and the recently discovered ActS, activate these amidases [20,37] has yet to be characterized, the mechanism that allosterically activates *C. difficile* CwlD almost certainly differs from the Ami family amidases.

Another question to be resolved is how CwlD binding to GerS affects GerS structure and thus stability. In the co-crystal structure, two GerS subunits intimately dimerize in a handshake like manner (**Figs 1 and 2**). Given the extensive interactions between the two monomers, GerS dimerization is likely critical for GerS to complex with CwlD. GerS dimerization may also help position CwlD's active site towards its peptidoglycan substrate, since GerS binds close to CwlD's N-terminal membrane-associated region in the co-crystal structure (**Fig 2**).

One way to address the importance of GerS dimerization would be by targeting amino acid residues in the cross-over arm. However, this may be challenging because those mutations could affect the stability of GerS itself. Interestingly, when the β-strand exchange between GerS subunits is eliminated by creating a GerS pseudo-monomer, GerS exhibits structural homology to the LolA/LolB family of proteins, which are responsible for the transport of lipoproteins across the periplasm of Gram-negative bacteria during assembly of the outer membrane [23]. The significance of this structural similarity remains unclear, since most LolA superfamily structures are monomeric. Intriguingly, a single mutation in LolA (F47E) has been reported to convert LolA from a monomer into a domain-swapped homodimer, and the homodimer binds LolA's interacting partner, LolC, more efficiently [25]. The equivalent residue in *C. difficile* GerS is an aspartate, raising the possibility that GerS could be converted into a monomer like LolA by mutating to phenylalanine. While future work will address the importance of GerS dimerization to its function, it is interesting to note that proteins with structural homology to GerS, LolA/LolB and RseB/MucB, bind lipids [24,38]. Thus, GerS may also bind lipids, which could impact its localization in sporulating cells.

While we cannot rule out the possibility that GerS regulates multiple aspects of CwlD function, e.g. localization, our results collectively suggest a new mechanism for activating amidases via stabilization of $Zn^{2+}$ co-factor binding and provide novel structural insight into how cell wall hydrolase activity can be regulated by protein binding partners. Indeed, stabilizing co-factor binding to enzymes could be a general strategy used by other peptidoglycan hydrolases. Also, given the diversity of hydrolases and regulators, distinct structures and regulatory mechanisms exist likely remain to be identified [17]. Finally, since both GerS lipoprotein and PdaA deacetylase impact *C. difficile* infection [9,12], our study suggests that inhibiting CwlD:GerS binding during spore formation could impair *C. difficile* spore germination and thus reduce disease transmission and recurrence.

## Material and methods

### Bacterial strains and growth conditions

*C. difficile* strain construction was performed using 630Δ*erm*Δ*cwlD*Δ*pyrE* and 630Δ*erm*Δ*gerS*Δ*pyrE* as the parental strains via *pyrE*-based allelic-coupled exchange (ACE ([30]). This system allows for single-copy complementation of the Δ*cwlD* and Δ*gerS* parental mutants, from an ectopic locus. The *C. difficile* strains used are listed in S2 Table. They were grown on brain

heart infusion media (BHIS) supplemented with taurocholate (TA, 0.1% wt/vol; 1.9 mM), thiamphenicol (10–15 μg /mL), kanamycin (50 μg/ml), cefoxitin (8 μg/ml) and L-cysteine (0.1% w/v; 8.25 mM) as needed. For ACE, *C. difficile* defined medium (CDDM) [39] was supplemented with 5-fluoroorotic acid (5-FOA) at 2 mg/ml and uracil at 5 μg/ml. Cultures were grown at 37˚C under anaerobic conditions using a gas mixture containing 85% $N_2$, 5% $CO_2$ and 10% $H_2$.

The *Escherichia coli* strains used for BL21(DE3)-based protein production and for HB101/ pRK24-based conjugations are listed in S2 Table. *E. coli* strains were grown at 37˚C with shaking at 225 rpm in Luria-Bertani broth (LB). The media was supplemented with chloramphenicol (20 μg/ml), ampicillin (50 μg/ml), or kanamycin (30 μg/ml) as indicated.

### *E. coli* strain construction

All primers used for cloning are listed in S3 Table. Details of *E. coli* strain construction are provided in S1 Text. Plasmid constructs were confirmed by sequencing using Genewiz and transformed into either the HB101/pRK24 conjugation strain (used with *C. difficile*) or BL21(DE3) expression strain.

### *C. difficile* strain construction and complementation

Allele-couple exchange (ACE) [30] was used to construct all complementation strains. Complementations were performed as previously described by conjugating HB101/ pRK24 carrying pMTL-YN1C plasmids into Δ*pyrE*-based strains [40] using allele-coupled exchange.

### Protein purification for recombinant *E. coli* analyses

Starter cultures were grown in 20 mL LB broth with 100 μg/mL ampicillin and, in some cases, 30 μg/mL of kanamycin. Terrific broth (TB) with ampicillin and kanamycin was inoculated with the starter culture (1:1000) and incubated for ~60 hr at 20˚C with 225 rpm shaking. The purifications were carried out in two steps: nickel bead-based affinity purification followed by size exclusion chromatography (SEC). Affinity purification was carried out as stated above: the cells were pelleted, resuspended in lysis buffer (500 mM NaCl, 50 mM Tris [pH 7.5], 15 mM imidazole, 10% [vol/vol] glycerol, 2 mM ß-mercaptoethanol), flash frozen in liquid nitrogen, thawed and then sonicated. The insoluble material was pelleted and the soluble fraction was incubated with Ni-NTA agarose beads (Qiagen) for 2 hours to capture the $His_6$ tagged proteins. Washes were carried out three times with low imidazole buffer (500mM NaCl, 10mM Tris-HCl pH 7.5, 10% (v/v) glycerol, 15 mM imidazole, 2 mM ß -mercaptoethanol) to decrease non-specific binding to the beads. Proteins were eluted using high-imidazole buffer (500 mM NaCl, 50 mM Tris [pH 7.5], 200 mM imidazole, 10% [vol/vol] glycerol, 2 mM ß-mercaptoethanol) after nutating the sample for 5 to 10 min. Pooled protein elutions were concentrated to 20 mg/ml or less in a gel filtration buffer consisting of 200 mM NaCl, 10 mM Tris HCl pH 7.5, 5% glycerol and 1 mM DTT. SEC was carried out using a Superdex 200 Increase 10/300 GL (GE Healthcare) column. Proteins were resolved on 14% SDS-PAGE gels, and the gels were stained using GelCode Blue according to the manufacturer's instructions (ThermoFisher Scientific). The purified proteins were concentrated, aliquoted, flash frozen, and stored at—20˚C for future analysis.

For EDTA and $Zn^{2+}Cl_2$ analyses, 50 uM of Ni-affinity purified protein complexes were incubated for 1h at 4˚C on gel filtration buffer with either 1 mM of EDTA or fivefold excess of $Zn^{2+}Cl_2$. Complex samples were then injected on the SEC column primed with gel filtration buffer containing 1 mM of EDTA or 50 uM of $Zn^{2+}Cl2$.

For elemental analyses, SEC-purified proteins were concentrated to ~ 6 mg/mL in 50 mM NaCl, 10 mM Tris HCl pH 7.5, 5% glycerol and 1 mM DTT, were diluted 10X in 0.1% HNO3 and analyzed by ICP-MS (8900, Agilent, Santa Clara, CA) in helium mode at the Dartmouth Trace Elemental Analyses Core.

## Protein purification for crystallography

*E. coli* B834 (DE3) methionine auxotrophic strain 2593 was used to co-affinity purify the soluble domain of GerS (GerS$_{\Delta22}$) with the His-tagged soluble domain of CwlD (CwlD$_{\Delta25}$). Minimal medium consists of M9 medium (1X M9 salts, 2 mM MgSO$_4$, 0.1mM CaCl$_2$ and 5 g/L (0.5%) glucose (dextrose), 40 mg/L of all amino acids except methionine, 40 mg/L Selenomethionine, 2 mg/L Thiamine and 2 mg/L biotin. Briefly, a starter culture was grown in minimal media supplemented with 5% LB overnight at 37˚C. Minimal media was then inoculated with started culture (1:100) and incubated at 37˚C to OD$_{600}$ of 0.6. Cells were induced with 200 uM isopropyl-β- D-1-thiogalactopyranoside (IPTG) for 5h and then harvested by centrifugation at 4000 rpm for 15 min. The complex was purified as described above. Protein purification was performed in the presence of the reducing agent TCEP (1 mM). The protein complex was then purified by SEC using a Superdex 200 Increase 10/300 GL (GE Healthcare) column. Finally, SEC fractions containing the CwlD:GerS complex were pooled, concentrated to 22 mg/mL, and flash frozen in liquid nitrogen.

## Crystallization and structure determination

Purified GerS, CwlD, and CwlD:GerS complex were evaluated for crystallization potential using dynamic light scattering using a DynaPro Protein Solutions DLS instrument (**S1 Fig**). Purified GerS showed multimodal polydispersity regardless of buffer conditions, in agreement with the gel filtration profile, and was prone to aggregation in DLS analyses (**S1 Fig**). Numerous crystallization matrix screens were performed with only phase separation observed for CwlD but several hits using PEG 3350 for the CwlD:GerS complex. The optimized condition for the complex contained 50 mM BisTris at pH 7.5 with PEG 3350 from 12–18%, 5% glycerol, and 0.25 M ammonium nitrate. Crystals were grown by hanging drop vapor diffusion at 18˚C with 30 mg/ml complex concentration mixed 1:1 with reservoir solution. Crystals were cryo-cooled in liquid nitrogen after soaking in crystallization reagent increased to 20% PEG 3350 and the addition of 14% ethylene glycol. The native complex and selenomethionine-substituted sample crystalized in the same condition. A derivative containing sodium iodide (NaI) was produced by soaking of native crystals with reagent containing 250 mM NaI for 10 minutes before cooling.

Data were collected at GM/CA Sector 23ID (Advanced Photon Source, Argonne National Laboratory) and data processed with the Proteum3 suite (Bruker AXS), Dials [41], and iMosflm [42]. Selenomethionine and NaI derivative data were collected at 12.659 kEv while the native data were collected at 12 kEv. Initial processing suggested space group P6$_1$22 or P6$_5$22 with one complex per asymmetric unit at a resolution of 2.4 Å (**S4 Table**). An initial solution was obtained by single anomalous dispersion (SAD) using the selenomethionine (SeMet/Se) incorporated crystal and the SAD routine from Crank2 [43] within CCP4 [44] and showed a preliminary C-alpha trace favoring the core beta sheet for GerS formed via a two-strand exchanged dimer. Data were subsequently integrated, scaled and merged with multiple programs and phasing trials repeated as Se-SAD using ShelX [45] within Crank2 to produce an initial solution using 6 Se sites (1 Se and 1 Zn$^{2+}$ from CwlD and 4 Se from GerS) CC$_{all}$/CC$_{weak}$ of 50.1/30.7 and CFOM of 80.9. Additional runs used the native, Se, NaI under multiple isomorphous replacement using Sharp/Autosharp (Global Phasing Limited) and produced near

identical solutions to the SAD and summarized in **S4 Table**. The $R_{merge}$ relative to the native data for MIR of the SeMet was 31.2% and for the Iodide 43.0% due to changes in cell edge lengths. Anomalous signal for all three data sets at the conserved $Zn^{2+}$ site for CwlD homologs was observed supporting the presence of the metal (**S2 Fig**). Automated iterative build and density modification cycles using Parrot and Buccaneer [44] typically achieved $R_{free}$ values near 30%. Manual building used Coot [46]with model refinement performed using Phenix with a final $R_{free}$ of 24.5%. Structural orthologs were identified and superimposed using the DALI protein structure comparison server [47]. All figures were prepared using PyMOL (Schrödinger Inc.).

## Plate-based sporulation

*C. difficile* strains were inoculated from glycerol stocks overnight onto BHIS plates containing taurocholate. Colonies arising from these plates were used to inoculate liquid BHIS cultures, which were grown to early stationary phase, back-diluted 1:50 into BHIS, and grown until the cultures reached an $OD_{600}$ between 0.35 and 0.75. Sporulation was induced on 70:30 agar plates as previously described [48] for 18–24 h, after which cells were analyzed by phase-contrast microscopy and harvested for western blot analyses.

## Immunoprecipitation analyses

Immunoprecipitations were performed on lysates prepared from cultures induced to sporulate on 70:30 plates for 22 h. The samples were processed as previously described [49]. Cultures from three 70:30 plates per strain were scraped into 3 x 1 mL of FLAG IP buffer (FIB: 50 mM Tris pH 7.5, 150 mM NaCl, 0.02% sodium azide, 1X Halt protease inhibitor (ThermoScientific). The cultures were transferred into tubes containing a ~300 μL of 0.1 mm zirconia/silica beads (BioSpec). The cultures were lysed using a FastPrep-24 (MP Biomedicals) for 4 x 60 s at 5.5 M/s, with 5 min cooling on ice between lysis intervals. The tubes were pelleted at 3,000 x g for 2 min to pellet beads, unbroken cells/spores, and insoluble material. After washing the tubes with additional FIB, the lysates were pooled, and NP-40 detergent was added to a final concentration of 0.1%, to solubilize membrane-associated proteins. Then, the FLAG-conjugated magnetic resin was added. This magnetic resin was generated by incubating Dynabead Protein G (ThermoScientific) with anti-FLAG antibodies (Sigma Aldrich) at room temperature followed by washing. The lysates were incubated with the anti-FLAG resin for 1 hour at room temperature with rotation. After washing the beads with FIB, FLAG-tagged proteins were eluted using FIB containing 0.1 mg/mL FLAG peptide (Sigma Aldrich). After cell lysis, NP-40 detergent was added to the FIB at a concentration of 0.1%. All immunoprecipitations were performed on three independent biological replicates.

## Western blot analysis

Samples for immunoblotting were prepared as previously described [50]. Briefly, sporulating cell pellets were resuspended in 100 μl of PBS, and 50 μl samples were freeze-thawed for three cycles and then resuspended in 100 μl EBB buffer (8 M urea, 2 M thiourea, 4% [wt/vol] SDS, 2% [vol/vol] ß-mercaptoethanol). *C. difficile* spores (~1 x $10^6$) were resuspended in EBB buffer, which can extract proteins in all layers of the spore, including the core. Both sporulating cells and spores were incubated at 95˚C for 20 min with vortex mixing. Samples were centrifuged for 5 min at 15,000 rpm, and 4 x sample buffer (40% [vol/vol] glycerol, 1 M Tris [pH 6.8], 20% [vol/vol] ß -mercaptoethanol, 8% [wt/vol] SDS, 0.04% [wt/vol] bromophenol blue) was added. Samples were incubated again at 95˚C for 5 to 10 min with vortex mixing followed by centrifugation for 5 min at 15,000 rpm. The samples were resolved by the use of 14% SDS-PAGE gels

and transferred to a Millipore Immobilon-FL polyvinylidene difluoride (PVDF) membrane. The membranes were blocked in Odyssey blocking buffer with 0.1% (vol/vol) Tween 20. Polyclonal rabbit anti-GerS and anti-CwlD [6] antibodies were used at a 1:1,000 dilution. Polyclonal mouse anti-SpoIVA [51] and mouse anti-SleC [40] antibodies were used at 1:2,500 dilution and 1:5,000 dilution, respectively. IRDye 680CW and 800CW infrared dye-conjugated secondary antibodies were used at 1:20,000 dilutions. Odyssey LiCor CLx was used to detect secondary antibody infrared fluorescence emissions.

## Spore purification

Sporulation was induced on 70:30 agar plates for 2 to 3 days as previously described [48]. Spores were washed 6 times in ice-cold water, incubated overnight in water on ice, treated with DNase I (New England Biolabs) at 37˚C for 60 min, and purified on a HistoDenz (Sigma-Aldrich) gradient. Phase-contrast microscopy was used to assess spore purity (>95% pure); the optical density of the spore stock was measured at $OD_{600}$; and spores were stored in water at 4˚C. Two 70:30 plates were used to induce sporulation for each strain; purified spores were resuspended in 200 μl, and the total optical density of this mixture was determined for each strain. At least three independent spore preparations were performed.

## Germination assay

Germination assays were performed as previously described [12]. Approximately $1 \times 10^7$ ($OD_{600}$ of 0.35) were resuspended in 100 μl of water, and 10 μl of this mixture was removed for 10-fold serial dilutions in PBS. The dilutions were plated on BHIS-TA, and colonies arising from germinated spores were enumerated at 20 h. Germination efficiencies were calculated by averaging the CFU produced by spores for a given strain relative to the number produced by wild-type spores for at least three biological replicates from two independent spore preparations. Statistical significance was determined by performing a one-way analysis of variance (ANOVA) on natural log-transformed data using Tukey's test.

## Germination OD600 kinetics assay and phase-contrast microscopy

For OD600 kinetics assays with taurocholate in 96-well plates, ~$1.6 \times 10^7$ spores (0.55 OD600 units) for each condition tested were resuspended in BHIS and 180 μL were aliquoted into two wells of a 96 well flat bottom tissue culture plate (Falcon) for each condition tested. The spores were exposed to 1% taurocholate, or water (untreated) in a final volume of 200 μL. The OD600 of the samples was measured every 3 minutes in a Synergy H1 microplate reader (Biotek) at 37˚C with constant shaking between readings. The change in OD600 over time was calculated as the ratio of the OD600 at each time point to the OD600 at time zero. These assays were performed on three independent spore preparations. Data shown are averages from three replicates.

For phase contrast microscopy, purified spores were resuspended to an $OD_{600}$ of 0.5 in BHIS ± 1% taurocholate and incubated aerobically for 90 min at 37˚C. Samples were spun down (5,000 x g, 10 min, 4˚C) washes with PBS and fixed with 3.7% paraformaldehyde for 15 min at RT. Following fixation, samples were washed with PBS and mounted on agarose pads.

## Supporting information

**S1 Fig. Comparison of the CwlD:GerS complex to solution behavior.** The left panel shows the two CwlD / two GerS complex within the crystal with dimensions along each axis shown in nm. Dynamic light scattering results are shown for purified CwlD, CwlD:GerS complex,

and GerS. The radius of the CwlD:GerS complex in solution of 5.6 nm agrees well with 5.8 nm as measured in the crystal. The 200 nm peak in the complex sample is excess GerS not in the complex as GerS in the absence of CwlD behaves multimodally with some aggregation. Purified GerS shows multiple peaks on SEC with the expected monomer peak redistributing back into multiple peaks upon storage. At low protein concentration (1 mg/mL) and high salt (500 mM NaCl), the DLS radius of 3.7 nm (diameter of 7.4 nm) is in agreement with the largest measurement of 7.8 nm for the GerS dimer structure when excluding the two CwlD molecules.
(TIF)

**S2 Fig. Heavy atom phasing of the CwlD:GerS complex structure.** Left shows the two CwlD (Blue tones) / 2 GerS (grey tones) complex. The yellow mesh shows the anomalous signal at 3 sigma for the selenomethionine derivative and the maroon the sodium iodide derivative from Autosharp (Global Phasing Limited). Middle panels show Se-SAD phases in blue and Se/NaI with native MIR phases in orange contoured at 1 sigma and the beta strand crossover between the two GerS protomers (solid box) with views 90° apart. Right panel (dotted box) shows the anomalous peaks near the zinc binding site of CwlD.
(TIF)

**S3 Fig. Size Exclusion Chromatography analyses of CwlD variants.** Purified His-tagged $CwlD_{WT}$, $CwlD_{H49A}$, and $CwlD_{E65A}$ were analyzed by size exclusion chromatography. mAU corresponds to the UV absorbance measurements ($A_{280}$) during the protein elution.
(TIF)

**S4 Fig. Coomassie stain of co-affinity purifications of His-tagged $CwlD_{WT}$ or $CwlD_{R169Q}$ with either $GerS_{WT}$ or $GerS_{D106A}$.** The indicated proteins were produced in *E. coli* and purified using $Ni^{2+}$- affinity resin. Cleared lysate (CL) and eluate (E) fractions were analyzed using Coomassie staining.
(TIF)

**S5 Fig. Germination efficiency of $CwlD_{FLAG}$ and $GerS_{FLAG}$ complementation strains.** (A) Change in the $OD_{600}$ in response to germinant of Δ*gerS* and Δ*cwlD* spores complemented with *gerS*$_{FLAG}$ and *cwlD*$_{FLAG}$ constructs, respectively, relative to spores complementated with native *gerS* or *cwlD*, respectively. Purified spores were resuspended in BHIS, and germination was induced by adding taurocholate (1% final concentration). The ratio of the $OD_{600}$ of each strain at a given time point relative to the $OD_{600}$ at time zero is plotted. The mean of three assays from 3 independent spore preparations are shown. Shading represents the standard deviation as the area between error bars for each time point measured. Statistical significance relative to wild-type was determined using two-way ANOVA and Tukey's test. **** $p < 0.0001$. (B) Spore germination efficiency of the strains relative complemented with FLAG-tagged vs. wild-type complementation constructs. The number of colony forming units (CFUs) produced by germinating spores is shown. Average of results from three independent spore preparations are shown along with the associated standard deviations. Statistical analyses relative to the wild-type were performed using one-way ANOVA and Tukey's test. ***, $p < 0.001$; **** $p < 0.0001$.
(TIF)

**S6 Fig. Spore germination of strains encoding FLAG-tagged $GerS_{D106R}$ variants monitored by phase-contrast microscopy.** (A) Purified spores were resuspended in BHIS ± germinant and incubated aerobically for 90 min at 37°C. At the indicated time points samples were fixed in paraformaldehyde and visualized using phase-contrast microscopy. Scale bar, 1 um. B) Percent spore germination over time as detected by phase-contrast microscopy. The percentage of

phase-bright spores of each strain at a given time point relative to the percentage at time zero is plotted. The mean of three assays from 2 independent spore preparations are shown. Shading represents the standard deviation as the area between error bars for each time point measured. Statistical significance relative to wild-type was determined using two-way ANOVA and Tukey's test. **** p < 0.0001.
(TIF)

**S1 Text.** *E. coli* **strain construction information.**
(DOCX)

**S2 Text. PDB X-ray Structure Validation Report.**
(PDF)

**S1 Table. Numerical data used in graphs.**
(XLSX)

**S2 Table. Strains and plasmids used in this study.**
(DOCX)

**S3 Table. Primers used in this study.**
(DOCX)

**S4 Table. Crystallographic data collection and structure refinement statistics for CwlD: GerS.**
(DOCX)

## Acknowledgments

We would like to thank B. Jackson at the Dartmouth Trace Element Analysis Core for help with the elemental analyses; N. Minton (U. Nottingham) for generously providing us with access to the 630Δ*erm*Δ*pyrE* strain and pMTL-YN1C and pMTL-YN3 plasmids for allele-coupled exchange (ACE).

## Author Contributions

**Conceptualization:** Carolina Alves Feliciano, Oscar R. Diaz.

**Data curation:** Brian E. Eckenroth, Oscar R. Diaz.

**Formal analysis:** Carolina Alves Feliciano, Brian E. Eckenroth, Oscar R. Diaz, Sylvie Doublié, Aimee Shen.

**Funding acquisition:** Carolina Alves Feliciano, Sylvie Doublié, Aimee Shen.

**Investigation:** Carolina Alves Feliciano, Brian E. Eckenroth, Aimee Shen.

**Methodology:** Carolina Alves Feliciano, Brian E. Eckenroth, Oscar R. Diaz, Sylvie Doublié, Aimee Shen.

**Project administration:** Sylvie Doublié, Aimee Shen.

**Resources:** Sylvie Doublié, Aimee Shen.

**Supervision:** Sylvie Doublié, Aimee Shen.

**Validation:** Carolina Alves Feliciano, Brian E. Eckenroth, Oscar R. Diaz, Aimee Shen.

**Visualization:** Carolina Alves Feliciano, Aimee Shen.

**Writing – original draft:** Carolina Alves Feliciano, Brian E. Eckenroth, Aimee Shen.

**Writing – review & editing:** Carolina Alves Feliciano, Oscar R. Diaz, Sylvie Doublié, Aimee Shen.

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
