## [Decision Letter · Decision Letter 0]

16 Aug 2021

Dear Dr Shen,

Thank you very much for submitting your Research Article entitled 'Allosteric activation of CwlD amidase activity by the GerS lipoprotein during Clostridioides difficile spore formation' to PLOS Genetics.

The manuscript was fully evaluated at the editorial level and by independent peer reviewers. The reviewers appreciated the attention to an important topic but identified some concerns that we ask you address in a revised manuscript

We therefore ask you to modify the manuscript according to the review recommendations. Your revisions should address the specific points made by each reviewer.

[LINK]

Yours sincerely,

Daniel B. Kearns

Associate Editor

PLOS Genetics

Lotte Søgaard-Andersen

Section Editor: Prokaryotic Genetics

PLOS Genetics

Reviewer's Responses to Questions

**Comments to the Authors:**

Reviewer #1: The manuscript by Feliciano et al. reports the first co-crystal structure of the soluble portions of a cell wall amidase (CwlD) with its cognate regulator (GerS). These proteins are involved in maturation of the spore peptidoglycan in C. difficile. The structure revealed that the GerS/CwlD complex displays a 2:2 stoichiometry and that GerS does not bindi near the active site of CwlD. Instead, in vitro experiments suggested that GerS regulates CwlD activity by modulating the ability of CwlD to bind Zn2+, and that disrupting Zn2+ binding via mutagenesis disrupts complex assembly. The authors demonstrated that CwlD variants which were incapable of binding Zn2+ exhibited a germination defect in vivo, likely due to the reduced protein stability of CwlD and GerS. Finally, guided by the structure, the authors identified two salt bridges that were critical for GerS binding to CwlD, and report that disrupting one of them retarded the speed of germination. The paper was well-written in a way that should be accessible to a general audience, and I imagine that the results will be of broad interest to bacterial cell and developmental biologists, enzymologists, and those who study bacterial pathogenesis. I have very few suggestions for improvement that the authors may or may not choose to incorporate.

Major comment:

1. Lines 249-251, Fig. 3E. The authors conclude that disrupting the Zn2+ binding site does not disrupt the stoichiometry of the complex (based on Coomassie staining in Fig. 3E), but to my eye it appears that the amount of CwlD in peak 1 is reduced relative to WT, especially with the H49A and the E65A variants. I realize the change in MW of the peak is annoyingly ~half of the MW of a single CwlD molecule, but perhaps the authors may consider raising the possibility that stoichiometry is affected, in addition to their preferred model that there is a conformational change upon disrupting Zn2+ binding? (optional)

Minor comments:

1. Figure 1. Please define in the figure legend the “25” next to the Coomassie-stained gels (presumably molecular weight marker?).

2. The bound divalent cation is referenced at different places in the paper as “zinc”, “Zn”, or “Zn2+”. For consistency, perhaps simply stick to “Zn2+” or “Zn2+ ion”?

Reviewer #2: This manuscript is an extremely well written, very clearly laid out, and comprehensive analysis of the genetics, biochemistry, and structure of the CwlD:GerS complex of Clostridium difficile. The authors show that GerS binds to CwlD and that the interaction allows conjugation of CwlD to a critical Zn active site cofactor. In turn, Zn binding preserves the CwlD:GerS interaction and stabilizes the GerS protein in vivo. They also establish an important contact salt bridge and even show that charge reversal on each protein partially reconnects the interaction. All told the work is a rather complete example of how peptidoglycan modification/hydrolase activity (CwlD) is allosterically regulated by another factor as well as how that interaction governs biochemistry and ultimately physiological role in spores. The data are beautiful. The only complaint I have is that the paper feels a tad long.

Line 125. One thing that I didn’t quite understand in the paper starts here. This line seems to suggest a 2:1 ratio of GerS:CwlD in SEC but later experiments DLS and 3D structure support a 2:2 ratio. Also, I didn’t understand how the “vice versa” would also satisfy the size of the complex. I don’t think this is a major issue, I just didn’t understand why the results were different or how it was being reconciled.

Line 160. I don’t think DLS has been defined at this point in the manuscript. Move acronym definition forward.

It is brought up a few times in the manuscript that homologs in Bacillus etc do not have a cognate GerS and do not require activation. That doesn’t preclude the possibility that GerS-lacking organisms do not yet encode regulators of CwlD, an inhibitor for example.

**Have all data underlying the figures and results presented in the manuscript been provided?**

Reviewer #1: Yes

Reviewer #2: None

PLOS authors have the option to publish the peer review history of their article (what does this mean?). If published, this will include your full peer review and any attached files.

Reviewer #1: No

Reviewer #2: No

---

## [Editor Report · Decision Letter 1]

23 Aug 2021

Dear Dr Shen,

We are pleased to inform you that your manuscript entitled "A lipoprotein allosterically activates the CwlD amidase during  Clostrioides difficile  spore formation" has been editorially accepted for publication in PLOS Genetics. Congratulations!

Yours sincerely,

Daniel B. Kearns

Associate Editor

PLOS Genetics

Lotte Søgaard-Andersen

Section Editor: Prokaryotic Genetics

PLOS Genetics

Comments from the reviewers (if applicable):

**Data Deposition**

http://datadryad.org/submit?journalID=pgenetics&manu=PGENETICS-D-21-00895R1

**Press Queries**

---

## [Editor Report · Acceptance letter]

17 Sep 2021

PGENETICS-D-21-00895R1 

A lipoprotein allosterically activates the CwlD amidase during *Clostridioides difficile* spore formation 

Dear Dr Shen, 

We are pleased to inform you that your manuscript entitled "A lipoprotein allosterically activates the CwlD amidase during *Clostridioides difficile* spore formation" has been formally accepted for publication in PLOS Genetics! Your manuscript is now with our production department and you will be notified of the publication date in due course.

With kind regards,

Andrea Szabo

PLOS Genetics

On behalf of:
